# Improving Data Efficiency for LLM Reinforcement Fine-tuning Through Difficulty-targeted Online Data Selection and Rollout Replay

**Yifan Sun**[†*]
UIUC

**Jingyan Shen**[*]
New York University

**Yibin Wang**[*]
UIUC

**Tianyu Chen**
University of Texas at Austin

**Zhendong Wang**
Microsoft

**Mingyuan Zhou**
University of Texas at Austin

**Huan Zhang**[†]
UIUC

## Abstract

Reinforcement learning (RL) has become an effective approach for fine-tuning large language models (LLMs), particularly to enhance their reasoning capabilities. However, RL fine-tuning remains highly resource-intensive, and existing work has largely overlooked the problem of data efficiency. In this paper, we propose two techniques to improve data efficiency in LLM RL fine-tuning: *difficulty-targeted online data selection* and *rollout replay*. We introduce the notion of adaptive difficulty to guide online data selection, prioritizing questions of moderate difficulty that are more likely to yield informative learning signals. To estimate adaptive difficulty efficiently, we develop an attention-based framework that requires rollouts for *only* a small reference set of questions. The adaptive difficulty of the remaining questions is then estimated based on their similarity to this set. To further reduce rollout cost, we introduce a rollout replay mechanism inspired by experience replay in traditional RL. This technique reuses recent rollouts, lowering per-step computation while maintaining stable updates. Experiments across 6 LLM-dataset combinations show that our method reduces RL fine-tuning time by $23\%$ to $62\%$ while reaching the same level of performance as the original GRPO algorithm. Our code repository is available at https://github.com/ASTRAL-Group/data-efficient-llm-rl/.

## 1 Introduction

Reinforcement learning (RL) has emerged as a promising and increasingly adopted paradigm for fine-tuning large language models (LLMs) toward stronger reasoning capabilities [10, 30, 15, 53]. Despite a steady stream of algorithmic improvements [51, 28, 1, 50], relatively little attention has been paid to improving the *data efficiency* of LLM RL fine-tuning. This gap is particularly concerning given that RL fine-tuning for LLMs is notoriously computationally expensive[1].

---

[*]Equal contribution. [†]Correspondence to Yifan Sun <yifan50@illinois.edu> and Huan Zhang <huan@huan-zhang.com>.

[1]For example, Luo et al. [30] report that training a relatively small 1.5B-parameter model on just 40K samples required over 3,800 A100 GPU hours—equivalent to approximately $4,500 in compute cost—even before scaling to larger models or longer training horizons.

39th Conference on Neural Information Processing Systems (NeurIPS 2025).

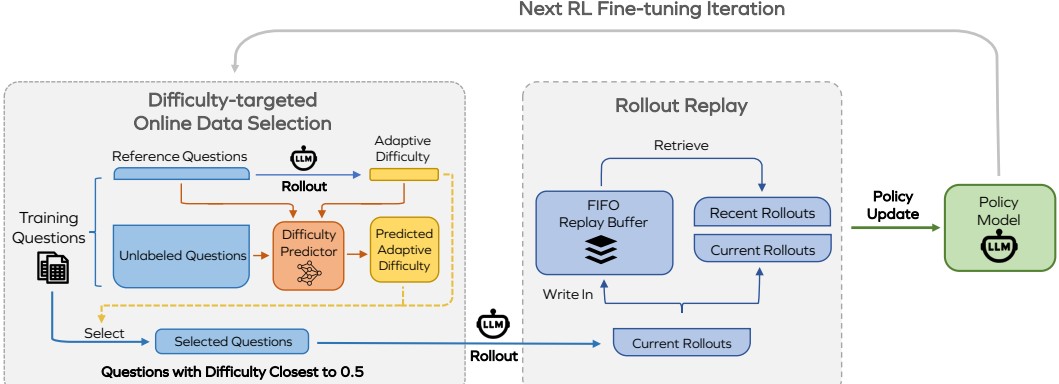

Figure 1: **Overview of our framework combining Difficulty-targeted Online Data Selection and Rollout Replay.** At each training step, the online data selection module selects training questions with adaptive difficulty near 0.5, requiring rollouts only on a small reference set (§4.1, §4.2). The rollout replay module combines current rollouts with retrieved recent rollouts from a FIFO buffer, and the current rollouts are stored into the buffer for future use (§4.3).

In this paper, we present two simple yet effective techniques to improve the data efficiency for LLM RL fine-tuning: **Difficulty-targeted Online Data Selection** and **Rollout Replay**. Our goal is to reduce both (1) the number of training steps required to match the performance of the original GRPO algorithm, and (2) the per-step computational cost.

$$\text{Total RL fine-tuning time} \quad = \quad \underbrace{\text{Number of training steps}}_{\text{Reduced by Difficulty-targeted Online Data Selection}} \quad \times \quad \underbrace{\text{Time per step}}_{\text{Reduced by Rollout Replay}}$$

**Difficulty-targeted Online Data Selection (DOTS)** In RL, tasks that are too easy or too difficult often provide limited learning signal [8, 17]. Moreover, since the policy evolves during training, it is crucial to adopt an online and adaptive mechanism for selecting informative data [34, 37]. To this end, we introduce the notion of *adaptive difficulty*, which measures how likely the current policy is to fail on a given question. At each training step, we prioritize questions of **moderate** adaptive difficulty, as these are most likely to yield meaningful learning signals.

However, computing adaptive difficulty exactly requires executing multiple rollouts per question, which is computationally expensive. To address this, we propose an *attention-based adaptive difficulty prediction framework* that efficiently estimates difficulty without generating full rollouts for all questions. At each training step, we generate rollouts only for a small reference set and compute their ground-truth adaptive difficulty. The difficulty of the remaining questions is estimated by comparing them to the reference set using similarity-based attention.

**Rollout Replay (RR)** To further reduce the cost of rollout generation, we introduce a simple rollout replay mechanism, motivated by experience replay in standard RL [7]. At each training step, we generate fewer new rollouts and reuse past rollouts from recent steps. A bounded First-In-First-Out (FIFO) buffer is maintained to store recent rollouts, from which we retrieve samples to complete each training batch. Although this makes GRPO slightly off-policy, our modified GRPO loss ensures stability, and thus RR effectively reduces per-step training time without degrading model performance.

**Our key contributions are summarized as follows:**

• We propose a novel attention-based adaptive difficulty prediction framework that efficiently estimates how likely a question will be answered incorrectly by the current policy, without requiring full rollouts for all questions.

• Guided by this difficulty prediction framework, we introduce an adaptive difficulty-targeted online data selection mechanism (DOTS) for RL fine-tuning, supported by theoretical justifications. DOTS prioritizes questions of moderate difficulty relative to the current policy, accelerating convergence.

• We develop a rollout replay (RR) mechanism that reuses recently generated rollouts. With a modified GRPO training loss, RR remains stable and effectively reduces per-step rollout cost.

• Extensive experiments on six LLM–dataset combinations show that our method reduces RL fine-tuning time by 23% to 62% while achieving the same performance as the original GRPO algorithm.

## 2 Related Work

**Online Data Selection**    Data selection seeks to accelerate training by focusing computation on the most informative examples [2]. A key limitation of static data selection methods is their assumption that the importance of samples remains fixed throughout training. Online methods instead periodically reselect data during training to reflect the model's evolving state [46, 52, 29, 16, 20]. Such adaptability is particularly important in RL, where non-stationary policy updates and environment dynamics necessitate continuous re-evaluation of data utility [34, 47, 36].

**Experience Replay**    On-policy algorithms such as Proximal Policy Optimization (PPO) [41] and Group Relative Policy Optimization (GRPO) [42] have become standard choices for online RL fine-tuning in LLM reasoning tasks [10]. However, their reliance on freshly collected rollouts for each policy update leads to substantial data inefficiency and computational overhead [27, 4]. Experience replay mitigates this by maintaining a fixed-size buffer of recent transitions collected by the policy. Instead of discarding data after a single use, the buffer enables multiple passes over past rollouts, thereby improving sample efficiency and stabilizing training [7, 54, 39].

## 3 Problem Setup

**GRPO**    We focus on the GRPO algorithm [42] with verifiable rewards. For each question $q$, a group of $G$ individual responses $\{o_i\}_{i=1}^{G}$ are sampled from the old policy $\pi_{\text{old}}$. The advantage of the $i$-th response is calculated by normalizing the group-level rewards $\{r_i\}_{i=1}^{G}$, where $r_i \in \{0, 1\}$:

$$\hat{A}_i = r_i - \text{mean}(\{r_i\}_{i=1}^{G}). \tag{1}$$

Compared to the original formulation proposed by [42], we remove the standard deviation normalization, as it has been shown to introduce bias into the optimization process [28]. Based on this, the GRPO objective can be formulated as:

$$\mathcal{J}_{\text{GRPO}}(\theta) = \mathbb{E}_{q \sim \mathcal{D},\ \{o_i\}_{i=1}^{G} \sim \pi_{\theta_{\text{old}}}(\cdot|q)}$$

$$\left[ \frac{1}{G} \sum_{i=1}^{G} \frac{1}{|o_i|} \sum_{t=1}^{|o_i|} \left( \min\left( r_{i,t}(\theta)\hat{A}_i,\ \text{clip}(r_{i,t}(\theta), 1-\epsilon, 1+\epsilon)\hat{A}_i \right) - \beta D_{\text{KL}}(\pi_\theta \parallel \pi_{\text{ref}}) \right) \right].$$

The first term represents a clipped policy update, where the ratio term $r_{i,t}(\theta) = \frac{\pi_\theta(o_{i,t}|q,o_{i,<t})}{\pi_{\theta_{\text{old}}}(o_{i,t}|q,o_{i,<t})}$ represents the probability ratio between the current and old policies. A KL penalty $D_{\text{KL}}(\pi_\theta \parallel \pi_{\text{ref}})$ is applied with respect to a fixed reference policy $\pi_{\text{ref}}$, weighted by a scalar coefficient $\beta$.

**Online Data Selection**    Let $\mathcal{D} = \{q_i\}_{i=1}^{N}$ denote the full dataset of $N$ questions. In standard GRPO, each policy update uses a batch of questions uniformly sampled from $\mathcal{D}$. However, not all questions contribute equally to learning progress. In particular, questions that are either too easy or too hard relative to the *current* policy's capability may yield weak gradient signals, slowing convergence.

To address this, we consider an **online data selection** setting [52]. At each step $t$, a batch $\mathcal{B}_t \subset \mathcal{D}$ of fixed size $B$ is selected based on the current policy $\pi_t$. Unlike static data pruning, this selection is repeated throughout training and adapts to the evolving policy. While more frequent selection allows better adaptation, it also increases computational overhead. In practice, when policy updates are relatively stable, it is often more efficient to perform data selection every $\mu$ (*e.g.,* 2,4,8...) steps, selecting a sequence of $\mu$ batches to be used in the subsequent updates [29, 44].

## 4 Method

Our proposed method is two-fold: (1) **Difficulty-targeted Online Data Selection**, which reduces the number of training steps needed to achieve the same performance as the original GRPO algorithm by prioritizing questions of moderate difficulty, and (2) **Rollout Replay**, which reduces the per-step computational cost by reusing recent rollouts. Full pseudocode is provided in Algorithm 1.

We propose using **adaptive difficulty** to guide online data selection. The adaptive difficulty of a question is defined with respect to the current policy and reflects how challenging the question is for

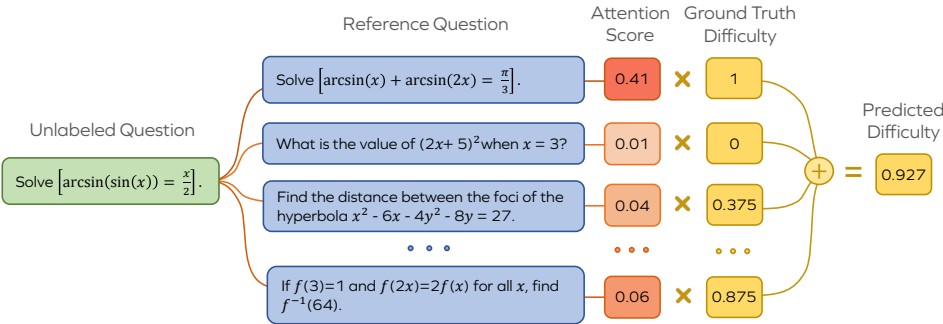

Figure 2: **Illustration of our attention-based adaptive difficulty prediction framework.** For each unlabeled question, we compute its embedding and attend to reference questions to obtain similarity scores. The predicted difficulty of the unlabeled question is obtained by computing an attention-weighted average, where similarities to reference questions serve as attention scores over their associated difficulties. In this example, the unlabeled question involves inverse trigonometric functions. The model assigns high attention to a reference question that tests a closely related concept and has a difficulty of 1.0. As a result, the predicted difficulty is also close to 1.0. All difficulty values shown correspond to *adaptive* difficulty scores computed at the same step.

the policy at the current stage of training. Formally, at step $t$, for each question $q$, we sample a group of $G$ responses $\{o_i^{(t)}\}_{i=1}^{G}$ from the current policy and obtain their corresponding rewards $\{r_i^{(t)}\}_{i=1}^{G}$. The adaptive difficulty at step $t$ is then computed as:

$$d_q^{(t)} = \frac{1}{G}\sum_{i=1}^{G}(1 - r_i^{(t)}). \tag{2}$$

This value represents the average failure rate under the current policy, with higher values indicating greater difficulty. Unlike static difficulty measures, adaptive difficulty evolves with the policy and provides a dynamic signal for selecting informative training samples.

**Challenge: How to estimate adaptive difficulty efficiently?**   A key challenge in using adaptive difficulty is that computing it requires executing multiple rollouts, which is one of the most expensive components in LLM RL fine-tuning[2]. This raises the question: *can we estimate adaptive difficulty efficiently without generating rollouts for all questions?* To address this, we propose a lightweight attention-based adaptive difficulty prediction framework that generates rollouts for only a small reference subset of questions. The adaptive difficulty of the remaining questions is then estimated by comparing them to reference questions with known difficulty values using similarity-based attention, thereby avoiding full rollouts. See Fig. 2 for an illustration.

### 4.1   Attention-based Adaptive Difficulty Prediction Framework

At each step $t$, given the full training dataset $\mathcal{D}$, we first sample a small subset of $K$ questions (e.g., 128 or 256) uniformly at random to form the **reference set** $\mathcal{D}_{\text{ref}}$. For each question in the reference set, we execute rollouts and compute its adaptive difficulty at step $t$, denoted by $\{d_i^{(t)}\}_{i=1}^{K}$ using Eq. 2.

For the remaining $N - K$ questions, we aim to estimate their adaptive difficulty *without performing rollouts*. To this end, we employ a lightweight embedding model $E_\theta$ to encode questions and capture similarity. We first compute the embeddings $\{z_i = E_\theta(q_i)\}_{i=1}^{K}$ for all reference questions. Denote $h$ as the embedding dimension. Then, for each unlabeled question $q$, we compute its embedding $z_q = E_\theta(q)$ and use similarity-weighted averaging to estimate its adaptive difficulty:

$$a_i = \frac{\exp(z_q^\top z_i/\sqrt{h})}{\sum_{j=1}^{K}\exp(z_q^\top z_j/\sqrt{h})}, \quad \hat{d}_q^{(t)} = \sum_{i=1}^{K}a_i d_i^{(t)}.$$

**Calibration**   To improve the prediction performance, we apply Platt scaling [35] that utilizes the information of mean and standard deviation of the reference set difficulties. Specifically, let

---

[2]For instance, generating rollouts for a batch of 512 samples with maximum sequence length 3072 takes 109.83 seconds on 8 L40S GPUs, nearly half of the total step time.

$\mu^{(t)} = \frac{1}{K} \sum_{i=1}^{K} d_i^{(t)}$ and $\sigma^{(t)} = \sqrt{\frac{1}{K} \sum_{i=1}^{K} (d_i^{(t)} - \mu^{(t)})^2}$ denote the mean and standard deviation of the reference difficulties at step $t$. These two statistics are passed through a lightweight MLP to produce scale and bias parameters $(w^{(t)}, b^{(t)}) = \text{MLP}([\mu^{(t)}, \sigma^{(t)}])$. We then apply a calibrated transformation to the predicted difficulty:

$$\hat{d}_{q,\text{cal}}^{(t)} = \sigma \left( w^{(t)} \cdot \left( \log \hat{d}_q^{(t)} - \log(1 - \hat{d}_q^{(t)}) \right) + b^{(t)} \right),$$

where $\sigma(\cdot)$ denotes the sigmoid function. The MLP is optimized using binary cross-entropy loss. Full training details can be found in §5.1 and Appendix C.1.

## 4.2 Adaptive Difficulty-targeted Online Data Selection

At each training step, with the adaptive difficulty prediction framework, we now efficiently obtain the adaptive difficulty for all questions in the training set. Inspired by prior work on goal curriculum in RL [8, 48], we prioritize questions whose predicted difficulty is closest to $0.5$.

This selection strategy selects questions that are neither too easy nor too hard for the current policy, as these are intuitively the most informative for learning. Moreover, in GRPO, when all sampled rewards for a question are either $0$ or $1$, the group-normalized advantage becomes identically zero, resulting in *no* gradient signal. By focusing on questions with predicted difficulty near $0.5$, we avoid such degenerate cases and ensure each update contributes meaningfully to policy gradients, thereby accelerating optimization convergence. We formalize this intuition in Theorem 1, which shows that the expected gradient magnitude is maximized when the reward success rate is $0.5$ (*i.e.*, the adaptive difficulty is also $0.5$). A complete proof is provided in Appendix B.

**Theorem 1** (Maximal Gradient Signal at 50% Success Rate). *Consider a single question $q$, where $G$ responses $\{o_i\}_{i=1}^{G}$ are sampled independently from the current policy $\pi_\theta(\cdot \mid q)$. Each response receives a binary reward $r_i \in \{0, 1\}$, sampled i.i.d. from a Bernoulli$(p)$ distribution, where $p$ represents the reward success rate. Define the group-relative advantage $\hat{A}_i$ as in Eq. 1. We consider the unclipped policy gradient estimator for this question without KL penalty $g = \sum_{i=1}^{G} \hat{A}_i \nabla_\theta \log \pi_\theta(o_i \mid q)$. Under mild assumptions on the reward and the likelihood gradients $\nabla_\theta \log \pi_\theta(o_i \mid q)$ (detailed in Appendix B), the expected squared norm of the gradient satisfies:*

$$\mathbb{E}[\|g\|^2] \propto p(1-p) \cdot (1 - 1/G),$$

*and is maximized when $p = 0.5$.*

**Discussion: Our dynamic selection mechanism *implicitly* promotes diversity.** As questions near the target difficulty are repeatedly selected and trained on, their predicted difficulty gradually deviates from $0.5$. They are then less likely to be sampled again, allowing other under-explored questions to enter the selection pool. This dynamic prevents overfitting to a small subset of questions and encourages broader coverage over time.

## 4.3 Rollout Replay

To further improve data efficiency, we aim to reduce the **time cost of each training step**. Since rollout generation is one of the most expensive components, we adopt a *rollout replay* mechanism, inspired by experience replay in traditional RL. Specifically, at each training step, we generate new rollouts for only a fraction $\delta B$ of the batch, where $\delta \in (0, 1]$, and fill the remaining $(1 - \delta)B$ samples using recent rollouts sampled from a FIFO replay buffer $\mathcal{D}_{\text{replay}}$ with capacity $C$.

However, naively reusing past rollouts introduces bias into the policy gradient estimation, as the data is no longer drawn from the current policy. This mismatch can lead to unstable training and performance degradation [31]. Inspired by off-policy variants of PPO [38], we propose a modified GRPO loss using importance sampling with respect to the behavior policy $\pi_{\theta_{\text{behavior}}}$ *under which the rollouts stored in the buffer were originally collected*:

$$\mathcal{J}_{\text{GRPO-RR}}(\theta) = \mathbb{E}_{q \sim \mathcal{D}, \ \{o_i\}_{i=1}^{G} \sim \pi_{\theta_{\text{behavior}}}(\cdot \mid q)}$$

$$\left[ \frac{1}{G} \sum_{i=1}^{G} \frac{1}{|o_i|} \sum_{t=1}^{|o_i|} \left( \min \left( \tilde{r}_{i,t}(\theta) A_i, \ \text{clip}(\tilde{r}_{i,t}(\theta), 1 - \epsilon, 1 + \epsilon) A_i \right) - \beta D_{\text{KL}}(\pi_\theta \| \pi_{\text{ref}}) \right) \right],$$

**Algorithm 1** GRPO with `DOTS` and `RR`

---

**Require:** Initial policy model $\pi_\theta$, reward model $r_\varphi$, training dataset $\mathcal{D}$, target difficulty $\alpha$, batch size $B$, total steps $T$, reference set size $K$, sampling temperature $\tau$, adaptive difficulty prediction framework DIFFPRED (§4.1), fresh rollout fraction $\delta \in (0, 1]$, buffer capacity $C$
1: Initialize replay buffer $\mathcal{R} \leftarrow \emptyset$
2: Set $\pi_{\theta_{old}} \leftarrow \pi_\theta$
3: **for** step $= 1, \ldots, T$ **do**
      *// Adaptive difficulty prediction*
4:      Sample reference set $\mathcal{D}_{ref} \subset \mathcal{D}$ uniformly at random, where $|\mathcal{D}_{ref}| = K$
5:      **for** each $q \in \mathcal{D}_{ref}$ **do**
6:          Generate $G$ outputs $\{o_i^q\}_{i=1}^G \sim \pi_{\theta_{old}}(\cdot \mid q)$
7:          Compute rewards $r_i^q = r_\varphi(o_i^q)$ for $i \in [G]$ and difficulty score $d_q = \frac{1}{G}\sum_{i=1}^G (1 - r_i^q)$
8:      **end for**
9:      Predict adaptive difficulty $\hat{d}_{q'} = \text{DIFFPRED}(\mathcal{D}_{ref}, \{d_q \mid q \in \mathcal{D}_{ref}\}, q')$ for all $q' \in \mathcal{D} \setminus \mathcal{D}_{ref}$
10:    Sample rollout batch $\mathcal{B}_{rollout}$ of size $\delta B$ from $\mathcal{D}$ according to:

$$P(q) = \frac{\exp\left(-|\hat{d}_q - \alpha|/\tau\right)}{\sum_{q' \in \mathcal{D}} \exp\left(-|\hat{d}_{q'} - \alpha|/\tau\right)}$$

11:    **for** each $q \in \mathcal{B}_{rollout}$ **do**
12:        Generate $G$ outputs $\{o_i^q\}_{i=1}^G \sim \pi_{\theta_{old}}(\cdot \mid q)$
13:        Compute rewards $r_i^q = r_\varphi(o_i^q)$ for $i \in [G]$ and group average reward $\bar{r}_q = \frac{1}{G}\sum_{i=1}^G r_i^q$
14:        Obtain advantages $\hat{A}_i^q$ and policy probabilities $\pi_{\theta_{old}}(o_i^q \mid q)$ for $i \in [G]$
15:    **end for**
      *// Rollout Replay and update*
16:    Sample $(1-\delta)B$ samples from buffer $\mathcal{R}$ to complete batch $\mathcal{B}$ if $|\mathcal{R}| \geq (1-\delta)B$
17:    Update policy $\pi_\theta$ using **modified** GRPO objective $\mathcal{J}_{\text{GRPO-RR}}$ on batch $\mathcal{B}$
      *// Store informative rollouts in buffer*
18:    Add $\left(q, \left\{(o_i^q, \hat{A}_i^q, \pi_{\theta_{old}}(o_i^q \mid q))\right\}_{i=1}^G\right)$ to $\mathcal{R}$ for $q \in \mathcal{B}_{rollout}$, $\bar{r}_q \notin \{0, 1\}$
19:    Remove the oldest samples from $\mathcal{R}$ until $|\mathcal{R}| \leq C$
20:    Set $\pi_{\theta_{old}} \leftarrow \pi_\theta$
21: **end for**

---

where $\tilde{r}_{i,t}(\theta) = \frac{\pi_\theta(o_{i,t} \mid q, o_{i,<t})}{\pi_{\theta_{behavior}}(o_{i,t} \mid q, o_{i,<t})}$. By appropriately controlling the buffer size $C$, we empirically demonstrate that rollout replay improves sample efficiency while maintaining training stability. For each newly generated rollout, if the group average reward is neither 0 nor 1 (*i.e.*, the sample yields a non-zero gradient signal), we store the question, its sampled rollouts, computed advantages, and policy probabilities into the buffer. When the buffer is full, the oldest samples are discarded.

## 5 Experiments

### 5.1 Experimental Setup

**LLMs and RL training datasets** We perform GRPO training on three model scales: Qwen2.5-Math-1.5B, Qwen2.5-3B, and Qwen2.5-Math-7B [49]. We adopt four open-source datasets for training: MATH [13], DeepScaleR-40K [30], Open-Reasoner-Zero-57K (ORZ) [15] and DeepMath-103K [12]. For MATH, we include all level 3–5 questions. For the other three datasets, we sample 8K to 10K subsets to construct the training pools. These datasets span diverse mathematical domains and difficulty levels. In total, we experiment with six LLM-training dataset combinations to assess the effectiveness of our framework.

**Implementation details for adaptive difficulty prediction framework** In practice, we observe that off-the-shelf pretrained embedding models struggle to capture fine-grained similarity between math questions. To address this, we freeze the Qwen2.5-Math-1.5B-Instruct backbone [49] and train a 3-layer MLP adapter with a calibration head, using binary cross-entropy loss. We fix the reference

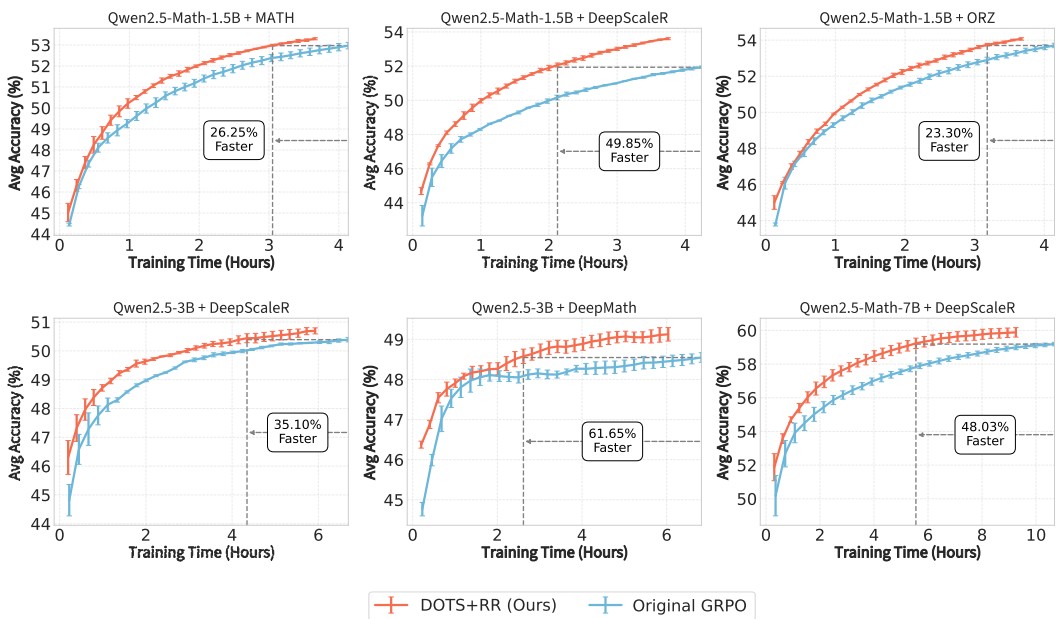

Figure 3: **Average accuracy curves of our method and original GRPO under various LLM–dataset combinations.** The curves show average performance aggregated over four benchmarks with exponential smoothing for visualization. The error bars represent 95% confidence intervals across 3 independent runs. Although both methods are trained for the same number of steps (60), our curve is shorter in duration because RR reduces the wall-clock time per step. Our method consistently outperforms the original GRPO throughout training and reduces the time required to match the original GRPO's final accuracy after 60 training steps by an average of 40.7%.

set size to 256. Additional implementation details are provided in Appendix C.1 and ablation results on key components are presented in Appendix E.1.

**Implementation details for RL training**  We employ the verl framework [43] to perform GRPO training. We use a batch size of 512 and a mini-batch size of 64 in verl's configuration, resulting in 8 gradient steps per training step. Across all experiments, we train for 60 training steps, yielding a total of 480 gradient steps. For each prompt, we generate 8 rollouts. The maximum rollout length is set to 3072 tokens for the Qwen2.5-Math series models (due to max position embedding limits) and 4096 tokens for Qwen2.5-3B. For reward computation, we use a simple rule-based function based solely on answer correctness, without incorporating any format-related signals. The 1.5B and 3B models are trained on 8 L40S GPUs, while the 7B model is trained on 8 A100 GPUs. For DOTS, data selection is performed every 2 steps. For RR, we choose the fresh rollout ratio $\delta$ as $0.5$ and buffer capacity $C \in \{256, 512\}$. All RL training hyperparameters are detailed in Appendix D.2.

**Evaluation**  We adopt the official Qwen2.5-Math evaluation implementation [49], setting the maximum generation length to 3072 tokens for Qwen2.5-Math series models and 4096 tokens for Qwen2.5-3B. Following [10, 30, 28], we evaluate RL model performance on standard mathematical reasoning benchmarks, including GSM8K [3], MATH500 [25], Minerva Math [22] and Olympiad-Bench [11]. Accuracy is measured using a sampling temperature of 0.6, top-p of 0.95, and the standard prompt template, consistent with [10]. We exclude benchmarks with very few questions, such as AIME 24 (30 questions) and AMC 23 (40 questions), as their small size results in high evaluation variance and unreliable performance comparisons on smaller models [14]. We report the final performance as the average accuracy across the four benchmarks to mitigate benchmark-specific variance. As the baseline, we use the original GRPO algorithm with uniform batch selection.

## 5.2  Main Results

The total training costs can be decomposed into two components: the number of steps required to reach a target performance and the average wall-clock time per step. Each training step involves

Table 1: **Percentage of training steps saved, per-step time saved, and total training time saved.** Results are averaged over four mathematical reasoning benchmarks and reported relative to the original GRPO baseline. All timing measurements are conducted on the same computational devices.

| Model | Dataset | Steps Saved (%) | Time Saved/Step (%) | Total Time Saved (%) |
|---|---|---|---|---|
| Qwen2.5-Math-1.5B | MATH | 16.67 | 11.71 | 26.25 |
| | DeepScaleR | 43.33 | 11.69 | 49.85 |
| | ORZ | 13.33 | 11.66 | 23.30 |
| Qwen2.5-3B | DeepScaleR | 26.67 | 11.52 | 35.10 |
| | DeepMath | 56.67 | 11.35 | 61.65 |
| Qwen2.5-Math-7B | DeepScaleR | 40.00 | 13.39 | 48.03 |

processing a fixed-size batch, consisting primarily of rollout generation and policy update. To ensure a fair comparison, each set of experiments is run on the same type and number of GPU devices.

**Our method reaches the same performance as the original GRPO with fewer steps.** Tab. 1 reports the number of training steps required by DOTS+RR to match the final performance of the original GRPO at 60 steps. Across all LLM–dataset combinations, our method consistently reaches the same performance with substantially fewer steps, achieving reductions ranging from 13.33% to 56.67%. These results demonstrate that DOTS significantly accelerates convergence by prioritizing informative training samples.

**Our method reduces per-step cost.** In our experiments, rollout generation accounts for approximately 47%, 46%, and 54% of the total per-step time for the 1.5B, 3B, and 7B models, respectively, with the remaining time primarily spent on policy updates[3]. By reducing the number of fresh rollouts per step, our RR strategy leads to a 11%–13% reduction in per-step training time, as shown in Tab. 1.

**Our method significantly reduces total training cost.** As shown in Fig. 3, DOTS+RR (orange) consistently outperforms the original GRPO (blue) throughout training, maintaining higher accuracy at almost every step. Across all six settings, DOTS+RR reduces total training time by an average of 40.7%, with the largest improvement observed on Qwen2.5-3B trained on DeepMath (61.65%).

## 5.3 Effectiveness of Adaptive Difficulty Prediction Framework

To better understand why our method accelerates training effectively, we examine whether the attention-based prediction framework can accurately estimate adaptive difficulty and consistently prioritize informative training signals throughout learning.

**The adaptive difficulty prediction aligns with evolving training dynamics.** To assess the fitness of online predictions, we collect ground-truth adaptive difficulty labels from training batches and compute the Pearson correlation

Table 2: **Average Pearson correlation ($\rho$) between predicted and ground-truth adaptive difficulties.** Reported as mean $\pm$ standard deviation over 60 training steps.

| Model | Dataset | $\rho$ |
|---|---|---|
| Qwen2.5-Math-1.5B | MATH | $0.7843 \pm 0.0243$ |
| | DeepScaleR | $0.7244 \pm 0.0318$ |
| | ORZ | $0.7153 \pm 0.0257$ |
| Qwen2.5-3B | DeepScaleR | $0.7789 \pm 0.0191$ |
| | DeepMath | $0.7029 \pm 0.0082$ |
| Qwen2.5-Math-7B | DeepScaleR | $0.7076 \pm 0.0195$ |

between these labels and the predicted difficulty scores. As shown in Tab. 2, our framework consistently achieves strong Pearson correlation ($\rho > 0.7$) across settings, demonstrating its ability to effectively track policy behavior throughout training. Additional qualitative examples are provided in Appendix C.2 to offer further insight into our attention-based prediction mechanism.

**Our prediction framework effectively filters out uninformative samples.** As discussed in §4.2, questions with adaptive difficulty values of 0 or 1 correspond to cases where all rollouts receive identical reward. In such cases, the group-normalized advantage becomes zero, yielding no gradient signal. We define *effective questions* as those with adaptive difficulty strictly between 0 and 1. As shown in Fig. 4, on average across all LLM-dataset combinations, DOTS selects 25.4% more effective

---

[3]In practice, for longer generation lengths, such as 8K and 16K, rollout time increases substantially, making it the dominant computational bottleneck. In such settings, our rollout replay mechanism can yield even greater wall-clock savings.

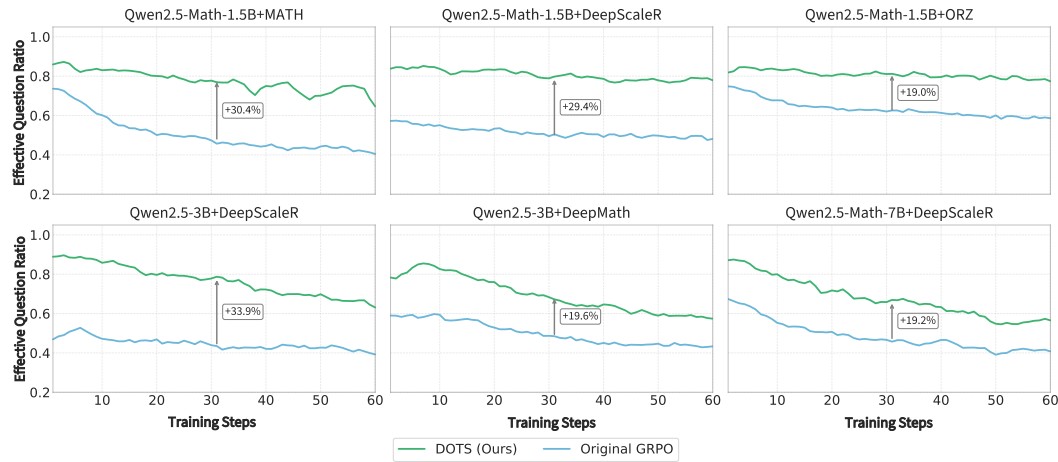

Figure 4: **Ratio of effective questions (*i.e.*, questions with adaptive difficulties strictly between 0 and 1) during training across various LLM-training dataset combinations.** Annotated percentages indicate the per-step increase in effective question ratio achieved by DOTS compared to original GRPO, averaged across the training process. Our adaptive prediction framework consistently selects more informative samples throughout training.

questions than the original GRPO, demonstrating a clear advantage in selecting more informative questions throughout training, thereby accelerating convergence.

**Our prediction framework incurs minimal computational overhead and scales efficiently to large datasets.** By caching question embeddings and using a lightweight encoder, our prediction framework remains highly efficient—processing 10K samples in just 1.71 seconds at deployment.

## 5.4 Analysis and Discussion

We further investigate three important questions: (**Q1**) What are the *individual* contributions of DOTS and RR to training efficiency? (**Q2**) How does DOTS compare to an online data selection method based on *external* difficulty labels? (**Q3**) Do DOTS and RR remain effective in *non-mathematical* domains?

**DOTS accelerates convergence, while RR reduces per-step cost.** As shown in Fig. 5(a), training guided by DOTS alone yields a steeper learning curve compared to original GRPO. Fig. 5(b) shows that incorporating RR further reduces training time by approximately 20% without sacrificing performance. These results show that DOTS and RR improve RL training efficiency in *complementary* ways.

**DOTS outperforms online data selection method based on external difficulty labels.** We compare DOTS with an online data selection baseline that relies on external difficulty annotations (*e.g.,* annotated by GPT-4o-mini), where training questions are selected at different stages based on static difficulty labels, gradually shifting from easier to harder questions over time.

Specifically, we use the DeepScaleR dataset and label each question with GPT-4o-mini, following the difficulty annotation prompt introduced in [30]. Each question is annotated 32 times, and the average score is used as its final difficulty. We then follow a staged curriculum: in the first third of training steps, batches are sampled from the easiest third of the dataset; in the middle third, from the medium-difficulty third; and in the final third, from the hardest third. To ensure a fair comparison of online data selection strategies, we compare this baseline with DOTS (without RR). As shown in Fig. 6(a), our DOTS method consistently outperforms this baseline on both Qwen2.5-Math-1.5B and Qwen2.5-3B. Moreover, such methods require expensive external labeling and offer limited adaptability, as they typically follow hand-crafted curricula that demand extensive manual design and tuning. In contrast, by leveraging adaptive difficulty, DOTS automatically adjusts to the model's learning progress without relying on external supervision, enabling more scalable and efficient training.

**DOTS and RR improve RL data efficiency beyond mathematics.** To further examine the generality of our approach beyond the math domain, we apply the full training and evaluation pipeline to the **science domain** using the curated SCP-25K dataset [26], which mostly contains advanced physics, chemistry, and biology questions. We adopt the Qwen2.5-3B model and train a new adaptive difficulty

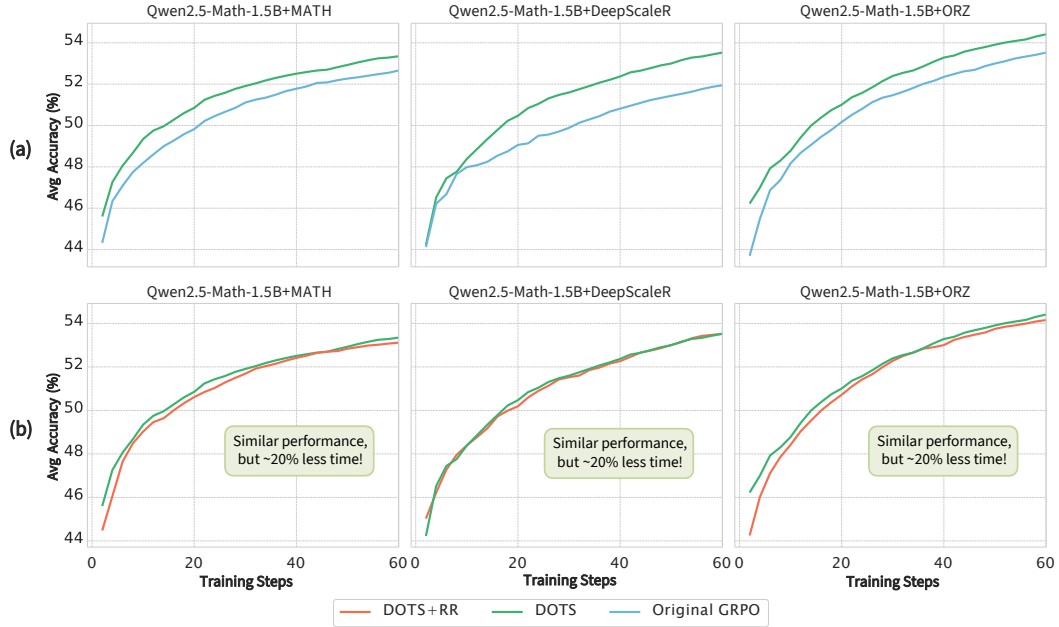

Figure 5: **Average accuracy curves of (a) DOTS vs. Original GRPO, and (b) DOTS+RR vs. DOTS on the Qwen2.5-Math-1.5B model.** The curves show average performance aggregated over four benchmarks with exponential smoothing. Note that the x-axis is the number of **steps** (rather than time). (a) DOTS consistently outperforms the original GRPO and leads to faster convergence. (b) Incorporating RR reduces training time by 20% while preserving the performance of DOTS.

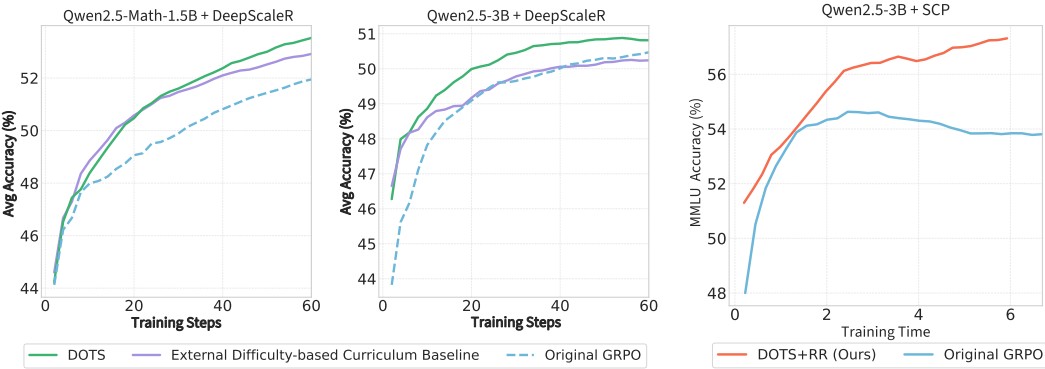

(a) **Comparison between DOTS (ours) and an external difficulty-based curriculum baseline.** The curves show average performance aggregated over four benchmarks with exponential smoothing for visualization. Note that the x-axis is the number of **steps** (rather than time). Our method consistently outperforms the baseline.

(b) **Results on the science subsets of MMLU using Qwen2.5-3B trained on the SCP-25K dataset.** Our method significantly outperforms original GRPO in this non-math domain.

Figure 6: **Comparison of DOTS with external difficulty-based curriculum baseline (left) and generalization to non-math domain (right).**

predictor, while keeping all other RL settings unchanged. We evaluate performance on the science subsets of MMLU, including questions in the fields of physics, chemistry, and biology. As reported in Fig. 6(b), our method continues to significantly improve RL data efficiency in this non-math domain, demonstrating its broader applicability.

## 6 Conclusion

In this paper, we propose two techniques to improve the data efficiency of LLM RL fine-tuning: Difficulty-targeted Online Data Selection and Rollout Replay. We hope these effective techniques will encourage future work to explore data-centric approaches to improving LLM RL fine-tuning.

## Acknowledgment

We would like to express our heartfelt thanks to Rayne Amami for helpful discussions and inputs.

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

# A  Discussions

## A.1  Limitations and Future Work

Our adaptive difficulty prediction framework currently relies on randomly sampling a reference set of $K$ questions at each selection step. While effective, the quality of the reference set can influence prediction performance. In principle, one could improve prediction performance by selecting a more diverse reference set that better covers the training set. Building on this idea, a natural extension is to fix a shared set of $K$ reference questions (with sufficient coverage) across training, re-evaluating their adaptive difficulty at each selection step.

Moreover, while we demonstrate the effectiveness of experience replay in the GRPO setting, our current strategy is relatively straightforward: we randomly replay rollouts associated with questions whose average reward across all rollouts is neither 0 nor 1. A promising direction for further improving efficiency is to incorporate more principled replay strategies, such as those inspired *prioritized experience replay* [40, 54].

Another potential extension of our method lies in the construction of input embeddings for difficulty prediction. Specifically, instead of relying solely on the question text, one could incorporate reference solutions to enrich the representation. Preliminary experiments suggest that including reference solutions can slightly improve the accuracy of adaptive difficulty prediction. However, this approach may have limited applicability in practice, as reference solutions are not available for all datasets (e.g., DeepScaler and ORZ).

Finally, we note that generating rollouts for the reference set can introduce nontrivial computational overhead, especially when the reference size is large. To mitigate this, we reuse rollouts from reference questions whose predicted difficulty is near 0.5, effectively incorporating them into training. This strategy reduces rollout generation cost by 4–12% per step while maintaining final performance.

## A.2  Extended Related Work

RL fine-tuning for LLMs (with verifiable rewards) has recently attracted significant attention, driven in part by the success of DeepSeek-R1 [10]. Compared to the original GRPO algorithm [42], recent work has proposed several algorithmic improvements: DAPO [51] introduces techniques such as clip-higher, dynamic sampling, token-level policy gradient loss, and overlong reward shaping, while Dr. GRPO [28] removes the length and standard deviation normalization terms to improve stability. Beyond these algorithmic enhancements, [45, 33] provide theoretical insights into GRPO, while [53, 50] conduct large-scale empirical studies across models, identifying key design choices that enable effective RL fine-tuning.

In contrast, relatively little attention has been paid to data-centric approaches, despite their demonstrated potential in other areas of LLM training [18, 19, 6]. LIMR [24] explores a static data selection strategy for RL fine-tuning by prioritizing samples based on their alignment with the policy's learning trajectory. However, it requires a full training run over the entire dataset beforehand, limiting its practicality. Our online data selection method `DOTS` is more efficient and applicable in realistic settings. In addition, prior work has not explored the use of rollout replay in LLM RL fine-tuning, which we show can further reduce training costs.

# B  Proofs

**Proof of Theorem 1.**  We restate Theorem 1 and provide a complete proof below.

**Theorem 1** (Maximal Gradient Signal at 50% Success Rate). *Consider a single question $q$, where $G$ responses $\{o_i\}_{i=1}^G$ are sampled independently from the current policy $\pi_\theta(\cdot \mid q)$. Each response receives a binary reward $r_i \in \{0, 1\}$, sampled i.i.d. from a Bernoulli($p$) distribution, where $p$ represents the reward success rate. Define the group-relative advantage $\hat{A}_i$ as in Eq. 1. We consider the unclipped policy gradient estimator for this question without KL penalty:*

$$g = \sum_{i=1}^{G} \hat{A}_i \nabla_\theta \log \pi_\theta(o_i \mid q).$$

*Under mild assumptions on the reward and the likelihood gradients $\nabla_\theta \log \pi_\theta(o_i \mid q)$, the expected squared norm of the gradient satisfies:*

$$\mathbb{E}[\|g\|^2] \propto p(1-p) \cdot (1 - 1/G),$$

*and is maximized when $p = 0.5$.*

*Proof.*  Let $r_i \in \{0, 1\}$ be the binary reward for response $o_i$, sampled i.i.d. from a Bernoulli($p$) distribution. Define the group-relative advantage as:

$$\hat{A}_i = r_i - \frac{1}{G} \sum_{j=1}^{G} r_j.$$

We aim to analyze the expected squared norm of the gradient estimator

$$g = \sum_{i=1}^{G} \hat{A}_i \nabla_\theta \log \pi_\theta(o_i \mid q).$$

Assume that the gradients $\nabla_\theta \log \pi_\theta(o_i \mid q)$ have bounded second moment:

$$\mathbb{E}[\|\nabla_\theta \log \pi_\theta(o_i \mid q)\|^2] \leq C < \infty.$$

We compute the full second moment of the gradient estimator, where the expectation is taken with respect to $\pi_\theta$:

$$\mathbb{E}[\|g\|^2] = \underbrace{\sum_{i,j=1}^{G} \mathbb{E}\left[\hat{A}_i \hat{A}_j\right] \cdot \mathbb{E}\left[\nabla_\theta \log \pi_\theta(o_i \mid q)^\top \nabla_\theta \log \pi_\theta(o_j \mid q)\right]}_{T_1}$$

$$+ \underbrace{\sum_{i,j=1}^{G} \mathrm{Cov}\left(\hat{A}_i \hat{A}_j, \ \nabla_\theta \log \pi_\theta(o_i \mid q)^\top \nabla_\theta \log \pi_\theta(o_j \mid q)\right)}_{T_2}.$$

We introduce a *weak-dependence assumption* that the correction term $T_2$ is negligible compared to the leading term $T_1$[4]:

$$\left|\frac{T_2}{T_1}\right| \ll 1.$$

Therefore, it suffices to focus our analysis on the leading term $T_1$.

By assumption, the log-likelihood gradients are zero-mean, independent, and identically distributed across $i$:

$$\mathbb{E}[\nabla_\theta \log \pi_\theta(o_i \mid q)^\top \nabla_\theta \log \pi_\theta(o_j \mid q)] = \begin{cases} V, & i = j, \\ 0, & i \neq j. \end{cases}$$

So,

$$\mathbb{E}[\|g\|^2] = V \cdot \sum_{i=1}^{G} \mathbb{E}[\hat{A}_i^2].$$

We now compute $\mathbb{E}[\hat{A}_i^2]$. Let $\bar{r} := \frac{1}{G} \sum_{j=1}^{G} r_j$, then:

$$\mathbb{E}[\hat{A}_i^2] = \mathbb{E}[(r_i - \bar{r})^2] = \mathrm{Var}(r_i - \bar{r}) = \mathrm{Var}(r_i) + \mathrm{Var}(\bar{r}) - 2\,\mathrm{Cov}(r_i, \bar{r}).$$

Since $r_i \sim \mathrm{Bernoulli}(p)$ and $r_j$ are i.i.d.,

$$\mathrm{Var}(r_i) = p(1-p), \quad \mathrm{Var}(\bar{r}) = \frac{p(1-p)}{G}, \quad \mathrm{Cov}(r_i, \bar{r}) = \frac{p(1-p)}{G}.$$

Substitute in:

$$\mathbb{E}[\hat{A}_i^2] = p(1-p) + \frac{p(1-p)}{G} - 2 \cdot \frac{p(1-p)}{G} = p(1-p)\left(1 - \frac{1}{G}\right).$$

Therefore,

$$\mathbb{E}[\|g\|^2] = V \cdot G \cdot p(1-p)\left(1 - \frac{1}{G}\right),$$

which is maximized when $p = 0.5$.

$\square$

---

[4]To support this assumption empirically, we compute the ratio $\left|\frac{T_2}{T_1}\right|$ on two LLM-dataset combinations. Specifically, we randomly sample 512 questions for each dataset, generate 8 rollouts per question, and evaluate the ratio. The results (mean $\pm$ standard deviation) are: Qwen2.5-Math-1.5B + MATH: $0.081 \pm 0.0065$, and Qwen2.5-3B + DeepMath: $0.081 \pm 0.0051$. These consistently low ratios empirically validate the weak-dependence assumption.

**Remark: Extension to Multi-component Rewards.** Theorem 1 focuses on binary rewards for simplicity, following standard practice in recent RLVR literature. Its core derivation—computing the second central moment of group-normalized rewards $\mathbb{E}[\hat{A}_i^2]$—extends naturally to more complex reward formulations.

For example, consider a reward composed of two independent components: a correctness term $c_i \sim \mathrm{Bern}(\alpha)$ and a format term $f_i \sim \mathrm{Bern}(\beta)$, where the total reward is $r_i = c_i + f_i$. Then,

$$\mathbb{E}[\hat{A}_i^2] = (\alpha(1-\alpha) + \beta(1-\beta))\left(1 - \frac{1}{G}\right),$$

which is maximized when both $\alpha = 0.5$ and $\beta = 0.5$. This demonstrates that our insight applies naturally to multi-component rewards and highlights the generality of the result.

## C Details of Adaptive Difficulty Prediction Framework

### C.1 Design and Implementation Details

The core of our adaptive difficulty prediction framework lies in obtaining proper embeddings to enable attention-based weighted prediction, as described in Section 4.1. To achieve this efficiently, we freeze the Qwen2.5-Math-1.5B-Instruct model as the backbone and augment it with a lightweight adapter and a calibration head.

The adapter is a GELU-activated MLP with three hidden layers, each containing 896 units and a dropout rate of 0.1. A LayerNorm is applied to the projection output to stabilize training. The calibration head is a two-layer MLP that takes the mean and standard deviation of reference set difficulties as input. The first output passes through a Softplus activation to yield the scale parameter $w^{(t)}$, while the second is transformed by a Tanh activation to produce a bounded bias term $b^{(t)}$.

We collect training data from a set of LLMs that are **disjoint** from our policy models. These include Qwen2.5-Instruct and Qwen2.5-Math-Instruct series [49], Eurus-2-7B-PRIME [5], Mathstral-7B-v0.1[5], DeepSeek-R1-Distill-Qwen-1.5B [10], DeepScaleR-1.5B-Preview [30], and Qwen2.5-7B-SimpleRL-Zoo [53]. For each model, we sample query questions and reference questions from math datasets and compute their adaptive difficulty as supervision labels. Specifically, each training instance consists of a query question $q$, a reference set with known difficulty scores $\{(q_i, d_i)\}_{i=1}^K$, and a ground-truth difficulty label $d_q$. Repeating this procedure across models yields the training dataset $\mathcal{D}_{\text{pred-train}}$.

We train the adapter and calibration head using the standard binary cross-entropy loss:

$$\mathcal{L}_{\text{BCE}} = -\frac{1}{|\mathcal{D}_{\text{pred-train}}|} \sum_{(q, \{(q_i, d_i)\}_{i=1}^K, d_q) \in \mathcal{D}_{\text{pred-train}}} \left[d_q \log \hat{d}_{q,\text{cal}} + (1-d_q)\log(1 - \hat{d}_{q,\text{cal}})\right],$$

where $\hat{d}_{q,\text{cal}}$ is the calibrated predicted difficulty for the query question.

### C.2 Qualitative Examples

Tab. 3 presents a qualitative example from the DeepScaler dataset using Qwen2.5-3B as the policy model, showing one unlabeled question along with the reference questions receiving the highest and lowest attention scores. The example demonstrates that our difficulty prediction framework assigns higher attention to reference questions that share key mathematical topics and structures (*e.g.*, rhombus, incircle), while down-weighting unrelated questions.

## D Implementation Details

### D.1 Training Datasets and Models

Our experiments involve three model sizes: Qwen2.5-Math-1.5B, Qwen2.5-3B, and Qwen2.5-Math-7B [49]. We adopt four open-source mathematical reasoning datasets for RL fine-tuning:

- **MATH** [13]: This dataset contains 12,500 competition-level problems from sources such as AMC and AIME, spanning seven mathematical subjects and five difficulty levels. Following [24, 53], we merge the train and test splits and retain only Level 3–5 questions. These are guaranteed to have no overlap with the MATH500 benchmark to prevent data contamination.
- **DeepScaleR-40K** [30]: A collection of approximately 40,000 curated mathematical problems from AMC (pre-2023), AIME (1984–2023), Omni-MATH [9], and Still [32]. Deduplication is performed

---

[5] https://huggingface.co/mistralai/Mathstral-7B-v0.1

Table 3: **Qualitative example illustrating the similarity-based attention mechanism in adaptive difficulty prediction.** The table shows one unlabeled question along with its top- and bottom-ranked reference questions by attention score. High-attention references (red) typically share similar concepts and difficulty with the target question (*e.g.*, rhombus and incircle geometry), while low-attention references (blue) diverge in topic and are substantially easier.

| **Data Source: DeepScaleR** | |
|---|---|
| **Unlabeled Question** | *[ground truth adaptive difficulty = 1.000, predicted difficulty = 0.907]* |

In the ⬚rhombus⬚ $ABCD$, point $Q$ divides side $BC$ in the ratio $1:3$ starting from vertex $B$, and point $E$ is the midpoint of side $AB$. It is known that the median $CF$ of triangle $CEQ$ is equal to $2\sqrt{2}$, and $EQ = \sqrt{2}$. Find the radius of the ⬚circle inscribed⬚ in rhombus $ABCD$.

| # | Attention Score | Adaptive Difficulty | Reference Question |
|---|---|---|---|
| 1 | 0.487 | 1.000 | Rhombus $ABCD$ has $\angle BAD < 90°$. There is a point $P$ on the ⬚incircle of the rhombus⬚ such that the distances from $P$ to the lines $DA$, $AB$, and $BC$ are 9, 5, and 16, respectively. Find the perimeter of $ABCD$. |
| 2 | 0.093 | 1.000 | ⬚Circle⬚ $\omega_1$ with radius 3 is ⬚inscribed⬚ in a strip $S$ having border lines $a$ and $b$. Circle $\omega_2$ within $S$ with radius 2 is tangent externally to circle $\omega_1$ and is also tangent to line $a$. Circle $\omega_3$ within $S$ is tangent externally to both circles $\omega_1$ and $\omega_2$, and is also tangent to line $b$. Compute the radius of circle $\omega_3$. |
| | | . . . | |
| 255 | 0.000 | 0.125 | A package of milk with a volume of 1 liter cost 60 rubles. Recently, for the purpose of economy, the manufacturer reduced the package volume to 0.9 liters and increased its price to 81 rubles. By what percentage did the manufacturer's revenue increase? |
| 256 | 0.000 | 0.125 | Given $\tan\left(\alpha - \frac{\pi}{4}\right) = 2$, find the value of $\sin\left(2\alpha - \frac{\pi}{4}\right)$. |

using embedding-based retrieval, and ungradable problems are filtered to ensure high-quality reward signals. We randomly sample 10,240 problems for training.

- **Open-Reasoner-Zero-57K (ORZ)** [15]: This dataset includes 57,000 high-quality reasoning problems sourced from AIME (up to 2023), AMC, MATH, Numina-MATH [23], and Tulu3 MATH [21]. Extensive cleaning via rule-based and LLM-based filters ensures evaluability and difficulty balance. We sample 8,192 problems for training.

- **DeepMath-103K** [12]: A large-scale dataset focused on high-difficulty mathematical problems, constructed with rigorous data decontamination procedures to support reliable benchmark evaluation. We sample 8,192 problems for training.

### D.2 RL Fine-tuning Details

Tab. 4 summarizes the hyperparameters used in our GRPO training. We adopt the same configuration across all experiments. Following [51, 15], we remove the KL regularization terms. For reward computation, we use a simple rule-based function based solely on answer correctness, without incorporating any format-related signals. Specifically, a reward of 1 is assigned for exact matches with the reference answer, and 0 otherwise. Answer matching is implemented using the Math-Verify library[6]. We adopt a standard chain-of-thought (CoT) prompt template, provided in Tab. 5.

### D.3 Implementation Details of `DOTS` and `RR`

We present the detailed hyperparameter settings of Algorithm 1 in Tab. 6. For `DOTS`, data selection is performed every two steps during RL fine-tuning.

---

[6]https://github.com/huggingface/Math-Verify

Table 4: **Detailed RL fine-tuning recipes.**

| | |
|---|---|
| Optimizer | AdamW |
| Total Batch Size | 512 |
| Mini Batch Size | 64 |
| Learning Rate | 1e-6 |
| LR Schedule | Constant |
| Weight Decay | 0 |
| Warm-up Ratio | 0 |
| Number of Training Steps | 60 |
| Number of Gradient Steps | 480 |
| Max Prompt Length | 1024 |
| Max Rollout Length | 3072/4096 |
| Number of Rollouts Per Prompt | 8 |
| Rollout Sampling Temperature | 0.6 |
| Rollout Sampling Top-p | 0.95 |
| GPU Hardware | 8x NVIDIA L40S/8x NVIDIA A100 |

Table 5: **Prompt template used for RL fine-tuning and evaluation.** The placeholder <question> is replaced with the actual mathematical question during fine-tuning and evaluation. Special tokens "<|im_start|>" and "<|im_end|>" are omitted for clarity.

**system**
Let's think step by step and output the final answer within \boxed{}.
**user**
<question>
**assistant**

## D.4 Evaluation Details

Consistent with RL fine-tuning, we use a sampling temperature of 0.6, top-p of 0.95, and the same prompt template. We evaluate model performance on four commonly-used mathematical reasoning benchmarks and report the average accuracy to mitigate benchmark-specific variance.

- **GSM8K** [3]: A test set of 1,319 grade school math word problems from the GSM8K dataset, requiring multi-step arithmetic reasoning.
- **MATH500** [25]: A widely used subset of the MATH test split [13]. These problems are excluded from our MATH training data.
- **Minerva Math** [22]: A set of 272 undergraduate-level science and math questions from MIT Open-CourseWare.
- **OlympiadBench** [11]: A benchmark of 675 problems from international math olympiads and physics contests.

We exclude benchmarks with very few questions, such as AIME 24 (30 questions) and AMC 23 (40 questions), as their limited size leads to high evaluation variance and unreliable performance comparisons for smaller models [14]. We further justify this exclusion by evaluating the original GRPO on AIME 24 across various LLM-dataset combinations. Specifically, each of the 30 AIME 24 questions is evaluated 8 times, and the average accuracy (avg@8) is computed at regular intervals during training. As shown in Table 7, the accuracy fluctuates considerably across training steps without a clear upward trend. This high variance across steps underscores the difficulty of obtaining reliable evaluation signals on such small-scale datasets, especially for smaller models with limited reasoning capacity.

## E  Additional Experimental Results

### E.1  Ablation Study on the Adaptive Difficulty Prediction Framework

**Off-the-shelf embeddings fail to capture difficulty structure.**   We evaluate a baseline that directly uses frozen embeddings from the Qwen2.5-Math-1.5B-Instruct model without any further training or calibration. In contrast, our framework incorporates trained adapter layers and a calibration head. As shown in Tab. 8, our framework consistently achieves significantly higher Pearson correlation with the ground-truth adaptive difficulty

Table 6: **Hyperparameters of** `DOTS` **and** `RR`.

| | |
|---|---|
| Target Difficulty $\alpha$ | 0.5 |
| Reference Set Size $K$ | 256 |
| Data Sampling Temperature $\tau$ | 1e-3 |
| Fresh Rollout Fraction $\delta$ | 0.5 |
| Buffer Capacity $C$ | 256/512 |

Table 7: **Accuracy of original GRPO on AIME 24 across training steps.** Each of the 30 questions is evaluated 8 times, and avg@8 accuracy is reported every 10 training steps. The results show high variance without clear trends, which limits evaluation reliability especially for smaller models.

| Steps | Qwen2.5-Math-1.5B + DeepScaler | Qwen2.5-3B + DeepScaler | Qwen2.5-Math-7B + DeepScaler |
|---|---|---|---|
| 10 | 10.42 | 7.50 | 20.42 |
| 20 | 15.83 | 6.25 | 24.17 |
| 30 | 15.83 | 8.75 | 24.17 |
| 40 | 11.67 | 9.17 | 23.33 |
| 50 | 16.67 | 7.50 | 25.42 |
| 60 | 13.75 | 5.42 | 20.00 |

across all settings. The poor performance of the off-the-shelf baseline highlights the necessity of further adapter layers and calibration for accurately predicting question difficulty.

Table 8: **Ablation study on training with adapter and calibration.** Comparison of average Pearson correlation ($\rho$) between predicted scores and ground-truth adaptive difficulties, reported as mean $\pm$ standard deviation over 60 training steps. Results show that training with adapter layers and calibration significantly improves prediction performance.

| Model | Dataset | Off-the-shelf Embedding | Our Method (With Adapter Layers + Calibration) |
|---|---|---|---|
| Qwen2.5-Math-1.5B | MATH | $0.2682 \pm 0.0207$ | $0.7843 \pm 0.0243$ |
| | DeepScaleR | $0.2064 \pm 0.0518$ | $0.7244 \pm 0.0318$ |
| | ORZ | $0.1598 \pm 0.0266$ | $0.7153 \pm 0.0257$ |
| Qwen2.5-3B | DeepScaleR | $0.2688 \pm 0.0369$ | $0.7789 \pm 0.0191$ |
| | DeepMath | $0.0671 \pm 0.0168$ | $0.7029 \pm 0.0082$ |
| Qwen2.5-Math-7B | DeepScaleR | $0.1983 \pm 0.0254$ | $0.7076 \pm 0.0195$ |

`DOTS` **is robust to the size of reference set.** We further investigate the impact of the reference set size $K$ in RL fine-tuning. Fig. 7 compares the performance of the original GRPO and `DOTS` under reference set sizes of 128 and 256, using Qwen2.5-Math-1.5B and Qwen2.5-3B on the DeepScaleR dataset. The results show that a reference set size of 128 yields RL performance comparable to that of 256. This indicates that `DOTS` is robust to smaller reference sets, enabling more efficient rollout collection without sacrificing RL fine-tuning quality.

## E.2 Additional Experiment with Extended Training Horizon

To further verify the stability of our findings, we extend training to 100 training steps (600 gradient steps) under two settings: Qwen2.5-Math-1.5B + DeepScaleR and Qwen2.5-3B + DeepMath. Notably, our method continues to outperform the original GRPO baseline.

## E.3 Additional Results under Different Evaluation Views

In the main text, Fig. 3 presents performance over wall-clock time, while Fig. 5 uses training steps. For completeness, we provide alternate versions: Fig. 9 shows the step-based view corresponding to Fig. 3, and Fig. 10 shows the time-based view for Fig. 5.

Across both views—training steps and wall-clock time—`DOTS+RR` and `DOTS` consistently demonstrate strong performance, confirming the robustness of our improvements regardless of presentation format. Interestingly, as shown in Fig. 10, although `DOTS` (without `RR`) incurs a small overhead from reference rollouts and difficulty prediction, it requires substantially fewer training steps to reach the same final accuracy as the original GRPO. As a result, the overall training time is often reduced despite the per-step overhead.

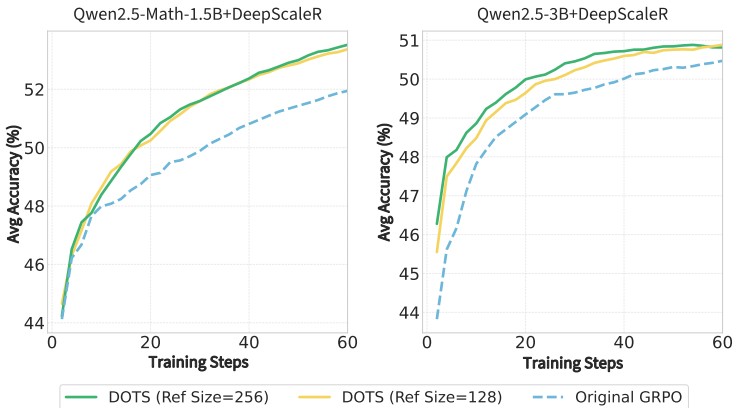

Figure 7: **Average accuracy curves of** `DOTS` **(Ref Size = 256),** `DOTS` **(Ref Size = 128), and Original GRPO on Qwen2.5-Math-1.5B and Qwen2.5-3B.** The curves show average performance aggregated over four benchmarks with exponential smoothing for visualization. Note that the x-axis is the number of **steps** (rather than time). The results show that a reference set size of 128 achieves performance comparable to that of 256, indicating the robustness of our method to smaller reference sets.

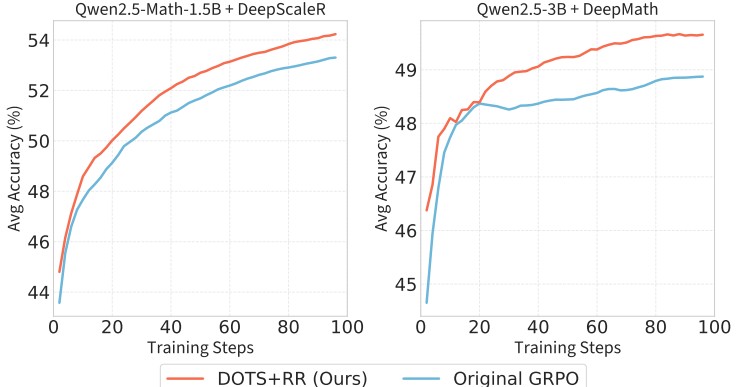

Figure 8: **Extended training to 100 training steps for two settings: Qwen2.5-Math-1.5B + DeepScaleR and Qwen2.5-3B + DeepMath**. Our method consistently outperforms the original GRPO baseline.

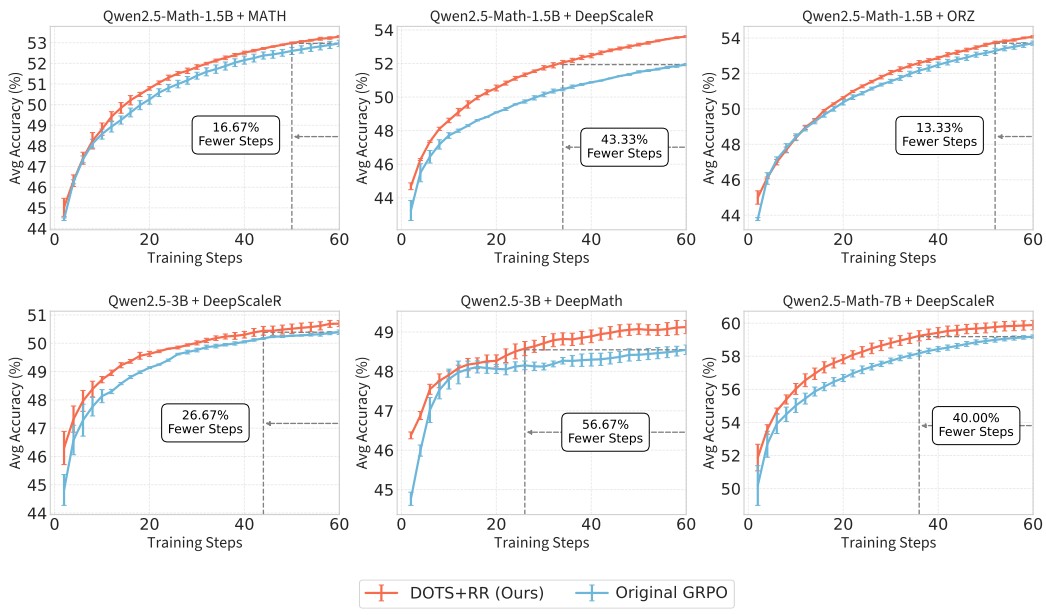

Figure 9: **Performance curves with training steps as x-axis.** Replot of Fig. 3 using training steps.

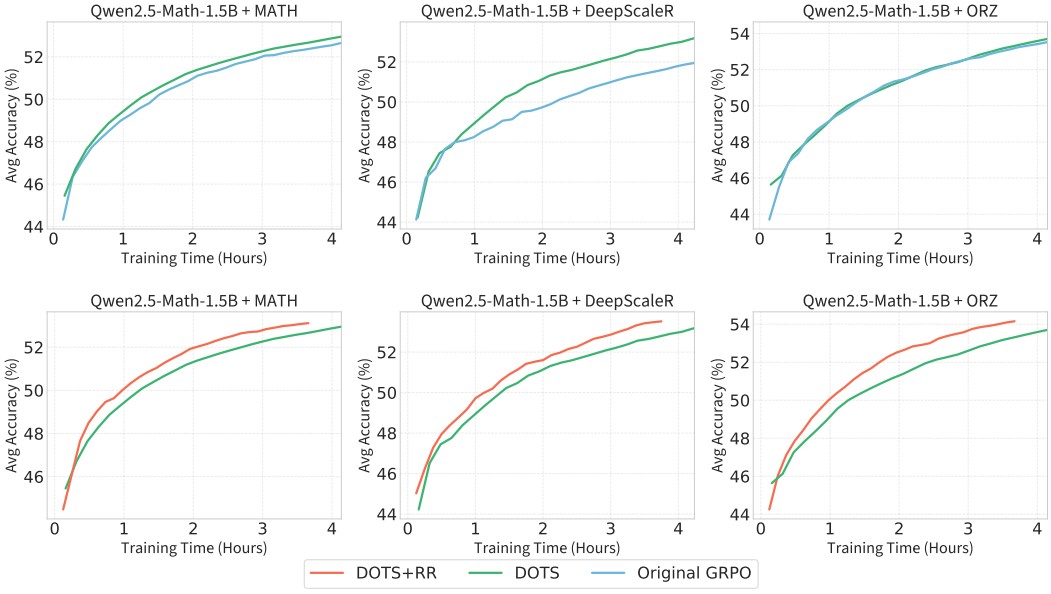

Figure 10: **Performance curves with wall-clock time as x-axis.** Replot of Fig. 5 using wall-clock time.

