# OpenReview forum: "Improving Data Efficiency for LLM Reinforcement Fine-tuning Through Difficulty-targeted Online Data Selection and Rollout Replay"
_NeurIPS.cc/2025/Conference — NeurIPS 2025 poster_

### Official Review · Reviewer_NvgW · 2025-06-27

**Clarity:** 4
**Significance:** 1
**Originality:** 2
**Rating:** 4
**Confidence:** 4

**Summary:**

In this paper, the authors address the computational inefficiency of reinforcement learning fine-tuning for large language models by introducing two techniques: Difficulty-targeted Online Data Selection (DOTS) and Rollout Replay (RR). DOTS employs an attention-based adaptive difficulty prediction framework that efficiently selects training questions with moderate difficulty, providing the most informative learning signals by generating rollouts for only a small reference set. Simultaneously, RR reduces computational costs by reusing recent rollouts and generating new rollouts for only a fraction of each training batch, using a modified loss function to maintain training stability. Experimentally validated across six LLM-dataset combinations, their approach demonstrates remarkable performance, reducing RL fine-tuning time while preserving the original GRPO algorithm's performance.

**Questions:**

1. The theorem is based only on binary rewards. How to generalize it to more complex reward scenarios?
2. Why did you choose a mathematical reasoning dataset?  Have you tried to verify it on non-mathematical fields?

**Ethical Concerns:**

["NO or VERY MINOR ethics concerns only"]

**Final Justification:**

The author's response to this paper addressed most of my concerns. Considering the contribution and application potential of this paper, I decided to improve my score.

**Limitations:**

1. Only validates mathematical reasoning datasets, generalization is unknown.
2. Difficulty selection may inadvertently reinforce training data bias.

**Paper Formatting Concerns:**

No major formatting issues.

**Quality:**

2

**Strengths And Weaknesses:**

Strengths:
1. Rigorous experimental methodology with comprehensive testing across multiple LLM models and datasets
2. Theoretically grounded approach, including a formal theorem (Theorem 1) explaining the rationale behind difficulty-based sample selection.
3. Demonstrates substantial time savings (25-65% reduction) across different model sizes and datasets.

Weaknesses:
1. No statistical significance tests due to high computational costs (acknowledged in the paper)
2. Pre-trained embedding models have difficulty capturing fine-grained similarities in mathematical problems. The paper alleviates this problem by fine-tuning an MLP adapter, but the adapter may overfit and its generalization ability to problems in different fields is questionable.

---

> ### Author Rebuttal · Authors · 2025-07-31
>
> We sincerely thank Reviewer NvgW for the thoughtful and constructive feedback. In response to the suggestions, we have conducted **additional experiments during the rebuttal period** on (1) adding **standard errors** for 3 LLM-dataset combinations to support statistical significance, and (2) evaluating our data-efficient RL training pipeline on a new **non-math** domain.
>
> Below, we address the questions and concerns point by point.
>
> ---
>
> > **W1: No statistical significance tests due to high computational costs**
>
> **Re.:** Thank you for the suggestion. We have added error bars for 3 LLM-dataset combinations by independently repeating both DOTS+RR and original GRPO three times. The results are shown in the table below and confirm that the improvements of our method over original GRPO are **statistically significant**, with **nearly all p-values below 0.05**.
>
> Specifically, we report the average accuracy (mean ± standard error) based on three independent runs. We also perform statistical testing using one-sided Welch’s t-tests. Due to the rebuttal policy, we cannot include plots, so we report each metric at 0.5-hour intervals in the tables below. Since DOTS+RR completes training earlier than original GRPO, we report results only up to the DOTS+RR completion point for comparison purposes.
>
> These additional runs represent our best effort within the limited rebuttal period and available compute resources. In the final version, we will extend this analysis to the remaining 3 combinations and increase the number of independent runs.
>
>
> **Table A: Average accuracy (mean ± standard error) of our method and original GRPO on Qwen2.5-Math-1.5B with DeepScaleR dataset.**
>
> | Time(h)    | DOTS+RR (Ours)    | Original GRPO    | p-value   |
> |--|--|--|--|
> | 0.50       | 48.11(±0.094)     | 47.18(±0.277)    | **0.033** |
> | 1.00       | 49.97(±0.128)     | 48.30(±0.057)    | **0.001** |
> | 1.50       | 51.12(±0.112)     | 49.29(±0.084)    | **0.000** |
> | 2.00       | 51.89(±0.155)     | 49.95(±0.126)    | **0.000**     |
> | 2.50       | 52.47(±0.132)     | 50.62(±0.097)    | **0.000**     |
> | 3.00       | 53.01(±0.096)     | 50.98(±0.027)    | **0.001**     |
> | 3.50       | 53.45(±0.034)     | 51.50(±0.064)    | **0.000**     |
> | 3.75       | 53.60(±0.058)     | 51.66(±0.053)    | **0.000**    |
>
>
>
> **Table B: Average accuracy (mean ± standard error) of our method and original GRPO on Qwen2.5-Math-1.5B with ORZ dataset.**
>
> | Time(h)    | DOTS+RR (Ours)    | Original GRPO    | p-value   |
> |--|--|--|--|
> | 0.50       | 47.68(±0.159)     | 47.79(±0.245)    | 0.636     |
> | 1.00       | 49.91(±0.045)     | 49.29(±0.131)    | **0.015**     |
> | 1.50       | 51.27(±0.080)     | 50.66(±0.130)    | **0.011**     |
> | 2.00       | 52.24(±0.133)     | 51.37(±0.080)    | **0.004**     |
> | 2.50       | 52.87(±0.092)     | 52.16(±0.166)    | **0.016**     |
> | 3.00       | 53.62(±0.086)     | 52.78(±0.148)    | **0.007**     |
> | 3.50       | 54.00(±0.046)     | 53.15(±0.141)    | **0.009**     |
> | 3.67       | 54.07(±0.067)     | 53.41(±0.139)    | **0.013**     |
>
>
> **Table C: Average accuracy (mean ± standard error) of our method and original GRPO on Qwen2.5-3B with DeepMath dataset.**
>
> | Time(h)    | DOTS+RR (Ours)    | Original GRPO    | p-value   |
> |--|--|--|--|
> |0.50| 46.88(±0.102)| 45.99(±0.141)    | **0.004**     |
> | 1.00| 47.91(±0.104)| 47.53(±0.228)    | 0.116     |
> | 1.50| 48.17(±0.155)| 48.06(±0.205)    | 0.346     |
> | 2.00| 48.27(±0.151)| 48.08(±0.133)    | 0.199     |
> | 2.50| 48.51(±0.189)| 48.05(±0.139)    | 0.064     |
> | 3.00| 48.71(±0.171) | 48.15(±0.110)    | **0.030**     |
> | 3.50| 48.82(±0.154)     | 48.12(±0.087)    | **0.013**     |
> | 4.00| 48.89(±0.161)     | 48.27(±0.135)    | **0.022**     |
> | 4.50| 48.98(±0.158)     | 48.30(±0.161)    | **0.020**     |
> | 5.00| 49.07(±0.132)     | 48.34(±0.148)    | **0.011**     |
> | 5.50| 49.05(±0.138)     | 48.42(±0.156)    | **0.020**     |
> | 6.00| 49.13(±0.167)     | 48.44(±0.141)    | **0.018**     |
> | 6.04| 49.13(±0.167)     | 48.46(±0.143)    | **0.020**     |
>
>
> > **Q2 & L1: Choice of mathematical reasoning dataset and generalization to non-math domains**
>
> **Re.:** We focused on mathematical reasoning datasets because math is a representative and important use case for LLM reasoning, and prior work on both algorithmic [1,2] and data-centric [3] improvements in LLM RL fine-tuning commonly adopts math datasets for training and evaluation.
>
> That said, inspired by your suggestion, we conduct an **additional experiment** in a non-math domain. Specifically, we apply our full pipeline, including both RL training and evaluation, to the **science domain** using the curated SCP-25K dataset [4], which contains advanced physics, chemistry, and biology questions. We use the Qwen2.5-3B model and train a new adaptive difficulty predictor. All other RL settings are kept unchanged.
>
> We evaluate performance on the science subsets of MMLU, and the results (reported in the table below) show that our method continues to **significantly improve RL data efficiency in this non-math domain**, supporting its **broader applicability**.
>
> **Table D: Average accuracy of DOTS+RR (ours) and original GRPO on Qwen2.5-3B with science subsets of MMLU.**
> |Step|DOTS+RR (Ours)|Original GRPO|
> |-|--|-|
> |10|**53.35**|53.28|
> |20|**55.36**|54.38|
> |30|**56.41**|54.44|
> |40|**56.48**|54.18|
> |50|**56.98**|53.84|
> |60|**57.31**| 53.80|
>
> > **W2: Generalization ability of the MLP adapter**
>
> **Re.:** We do not require the MLP adapter to transfer across domains. Instead, for each domain, we can train a separate adapter, which is both **justified and practical**.
>
> - **Justification:** This adapter is trained **specifically for the data domain used in the subsequent LLM RL fine-tuning**, which is **known in advance**. **If data from a particular domain will be used during RL fine-tuning, it is naturally included in the adapter training data.** Therefore, there is no need to generalize to out-of-domain data. This setting is fundamentally different from standard supervised learning, where models are expected to perform well on unseen distributions.
>
> - **Practicality:** The training cost is low, taking about 11.5 minutes on a single NVIDIA A100 GPU. This makes per-domain retraining entirely manageable in practice.
>
>
> > **Q1: Generalization of Theorem 1 to complex rewards**
>
> **Re.:** Theorem 1 uses binary rewards for simplicity, a standard choice in recent literature on RLVR (Reinforcement Learning with Verifiable Rewards). Its key step involves computing the second central moment of the group-normalized rewards $\mathbb{E}[\hat{A}_i^2]$, which can be easily recomputed under more complex reward formulations.
>
> For example, if the reward consists of two independent components such as a correctness reward $c_i \sim \text{Bern}(\alpha)$ and a format reward $f_i \sim \text{Bern}(\beta)$, with $r_i = c_i + f_i$, then we have $\mathbb{E}[\hat{A}_i^2] = (\alpha(1-\alpha) + \beta(1-\beta))(1 - 1/G)$, which is maximized when both $\alpha = 0.5$ and $\beta = 0.5$. This shows that our insight generalizes naturally to multi-component reward settings.
>
> > **L2: Potential bias due to difficulty-based selection**
>
> **Re.:** If the concern refers to the possibility that our difficulty prediction framework consistently favors a particular type of question (e.g., from certain domains or with certain formats), we believe this is **not a significant issue** in our setup.
>
> Since our method selects questions whose predicted adaptive difficulty under the *current* policy is close to 0.5, those that have been repeatedly selected and trained on will become easier and thus gradually fall outside the 0.5 region. As a result, they are less likely to be selected again, allowing other question types to enter the selection pool over time. This dynamic naturally promotes broader coverage during training.
>
> We will include a relevant discussion in the revised version.
>
> ---
>
> [1] Yu, Qiying, Zheng Zhang, Ruofei Zhu, Yufeng Yuan, Xiaochen Zuo, Yu Yue, Weinan Dai et al. "Dapo: An open-source llm reinforcement learning system at scale." arXiv preprint arXiv:2503.14476 (2025).
>
> [2] Lin, Zhihang, Mingbao Lin, Yuan Xie, and Rongrong Ji. "Cppo: Accelerating the training of group relative policy optimization-based reasoning models." arXiv preprint arXiv:2503.22342 (2025).
>
> [3] Li, Xuefeng, Haoyang Zou, and Pengfei Liu. "Limr: Less is more for rl scaling." arXiv preprint arXiv:2502.11886 (2025).
>
> [4] Liu, Mingjie, et al. "Prorl: Prolonged reinforcement learning expands reasoning boundaries in large language models." arXiv preprint arXiv:2505.24864 (2025).
>
> [5] Zeng, Thomas, Shuibai Zhang, Shutong Wu, Christian Classen, Daewon Chae, Ethan Ewer, Minjae Lee et al. "VersaPRM: Multi-Domain Process Reward Model via Synthetic Reasoning Data." In Forty-second International Conference on Machine Learning.
>
> [6] Namgoong, Hyuk, Jeesu Jung, Sangkeun Jung, and Yoonhyung Roh. "Exploring domain robust lightweight reward models based on router mechanism." arXiv preprint arXiv:2407.17546 (2024).

---

> > ### Comment · Reviewer_NvgW · 2025-08-04
> >
> > Thanks for the response, which addressed most of my concerns. I will raise my score.

---

> > > ### Author Response · Authors · 2025-08-04
> > > **Thank you!**
> > >
> > > Thank you for taking the time to reevaluate our work and for your thoughtful feedback! We will include these additional experimental results in the final version, which further strengthen our paper.
> > >
> > > If you have any further questions or suggestions, please feel free to let us know—we strive to consistently improve the quality and clarity of our paper.

---

### Official Review · Reviewer_puy2 · 2025-06-28

**Clarity:** 3
**Significance:** 3
**Originality:** 3
**Rating:** 4
**Confidence:** 4

**Summary:**

This paper aims to improve the data efficiency of reinforcement learning for LLM post-training. The authors recognized problems that not all training samples are used effectively to provide useful learning signals, and data from past iterations are simply discarded. Therefore, the authors proposed to techniques, difficulty-targeted online data selection (DOTS) and rollout replay (RR) to reduce the number of training steps and the computational costs per step. Specifically, DOTS introduces an “adaptive difficulty” metric that selects data of moderate difficulty to maximize the learning signal for each training step, while RR lowers the cost by reusing rollouts from recent training steps. With these two techniques combined, the paper demonstrates that the method reduces the RL training time by 25% to 65% percent without sacrificing performance.

**Questions:**

1. It is unclear from the paper whether the reference set’s rollouts are used in the training process other than the adaptive difficulty prediction.
2. Relatedly, when the reference set is 256 and the training batch size is 512, substantial time is spent on the rollout of the reference set. Have you considered using those samples for training (especially when their difficulty is around 0.5), for replay, or are they just discarded?
3. While there are high-level computational time comparisons, it is unclear how much time was taken for the additional difficulty prediction step. Can the authors present a breakdown of time for the overall process? Also, will the efficiency advantage (wall-clock time wise) still be present when there is only POTS (no RR)?
4. The experimental results lack credibility and reproducibility. There is only one run reported, without error bars. The experimental run is also terminated early - the convergence behavior is not seen in the plots.
5. Another questionable presentation of the result is that the ablation results in Figure 5 are shown on training step, while Figure 3 is shown on GPU time. I would like to see both as the x axis, for both overall result and ablation to assess reliably.
6. Though AIME results are excluded, they could have been reported separately because it could be possible that the designed online data selection method does not help with learning extremely hard questions like AIME.

**Ethical Concerns:**

["NO or VERY MINOR ethics concerns only"]

**Final Justification:**

The authors provided additional results during the short rebuttal period. My questions are answered. Though the improvements are marginal in some experiments, I feel the paper is above the threshold for acceptance. I will raise my score.

**Limitations:**

Please discuss the limitations in the main paper instead of the appendix B.

**Quality:**

2

**Strengths And Weaknesses:**

Strengths:

1. Recognizing the data efficiency in RL as a major problem to address, this is also a timely contribution.
2. The results do show the great performance of the proposed method in terms of efficiency gain.
3. The proposed method is shown to greatly improve the ratio of effective questions in the training, as shown in Figure 4.

Weaknesses:

1. The empirical results could lack credibility due to the limited number of runs, and some additional issues described in the Questions below.

---

> ### Author Rebuttal · Authors · 2025-07-31
>
> We sincerely thank Reviewer puy2 for the thoughtful and constructive feedback. In response to the suggestions, we have conducted **additional experiments during the rebuttal period** on (1) adding **standard errors** for 3 LLM-dataset combinations to support statistical significance, (2) **extending the training horizon** to observe convergence behavior, and (3) evaluating on the **AIME benchmark**.
>
> Below, we address the questions and concerns point by point.
>
> > **W1 & Q4 Part 1: No error bars**
>
> **Re.:** Thank you for the suggestion. We have added error bars for 3 LLM-dataset combinations by independently repeating both DOTS+RR and original GRPO three times. The results confirm that the improvements of our method over original GRPO are **statistically significant**, with **nearly all p-values below 0.05**.
>
> Due to space limitations, detailed results are provided in our response to Reviewer NvgW, W1.
>
> > **Q4 Part 2: Extended training to observe convergence behavior**
>
> **Re.:** Due to resource constraints, we initially limited all runs to 60 steps. Following your suggestion, we extended training to **100 steps** for two settings: Qwen2.5-Math-1.5B+DeepScaleR and Qwen2.5-3B+DeepMath. The updated results are shown in the table below and remain consistent with our original conclusions.
>
> **Table A: Average accuracy (%) of DOTS+RR (Ours) and original GRPO over 100 training steps on Qwen2.5-Math-1.5B with DeepScaleR and Qwen2.5-3B with DeepMath datasets.**
>
> |Steps|Qwen2.5-Math-1.5B+DeepScaleR||Qwen2.5-3B+DeepMath||
> |-|-|-|-|-|
> ||DOTS+RR|Original GRPO|DOTS+RR|Original GRPO|
> |10|**48.59** |47.67|**48.10**|47.74|
> |20|**50.02** |49.13|**48.39**|48.37|
> |30|**51.19** |50.36|**48.88**|48.26|
> |40|**52.09** |51.13|**49.06**|48.37|
> |50|**52.70** |51.69|**49.24**|48.45|
> |60|**53.14** |52.19|**49.38**|48.57|
> |70|**53.49** |52.62|**49.51**|48.62|
> |80|**53.84** |52.91|**49.63**|48.79|
> |90|**54.07** |53.15|**49.64**|48.85|
> |100|**54.31**|53.29|**49.66**|48.87|
>
> > **Q1 & Q2: Reuse of reference set rollouts**
>
> **Re.:** This is an insightful point. In our current experiments, rollouts from the reference set are **not reused** for training.
>
> Inspired by your suggestion, we conducted **additional experiments** to assess the benefit of reusing reference rollouts for training when their predicted difficulty is near 0.5. Approximately 10% of the reference questions per step fall into this moderate-difficulty range (see Table B below). Reusing the rollouts associated with these questions in RL training can **reduce overall rollout generation time by 4–12%** per step, while maintaining the same final policy performance. **We find this strategy effective and will adopt it in the revised version.**
>
> Relatedly, to **clarify the overhead**, in our experiments, data selection is conducted **every 2 steps** (stated in L185), so after amortization, **only 128 (instead of 256)** reference questions are needed to do rollouts per step.
>
> **Table B: Proportion of questions with adaptive difficulty=0.5 in reference sets at different training steps.**
> |Model+Dataset|Step 0|Step 20|Step 40|Step 60|
> |-|-|-|-|-|
> |Qwen2.5-Math-1.5B+DeepScaleR|0.11| 0.08|0.07|0.04|
> |Qwen2.5-Math-1.5B+ORZ|0.11|0.13|0.08|0.12|
> |Qwen2.5-3B+DeepMath|0.12|0.06|0.04|0.04|
>
> > **Q3 Part 1: Detailed time breakdown of the overall process**
>
> **Re.:** Thank you for your question. We provide a detailed breakdown of the time cost of DOTS+RR (Ours) on the Qwen2.5-Math-1.5B and Qwen2.5-3B models trained on DeepScaleR, averaged over 60 steps. The experiments are conducted on 8x NVIDIA L40S GPUs. The overall process consists of four components: (1) **Reference rollout time**, (2) **Adaptive difficulty prediction time**, (3) **Selected question rollout time**, and (4) **Actor update time**.
>
> Below is the breakdown of the average time (in seconds) amortized per step. We highlight three key observations:
>
> (1) The cost of adaptive difficulty prediction is negligible.
>
> (2) The selected question rollout time is relatively short due to the use of replay in DOTS+RR.
>
> (3) As mentioned in our response to Q1&Q2, reference rollouts whose predicted difficulty is close to 0.5 can be reused for training, thereby further reducing the cost of selected question rollouts.
>
> **Table C: Breakdown of the average time (in seconds) amortized per step of DOTS+RR.** We note that data selection is performed once every two steps (as stated in L185), so reference rollouts are only generated at those steps, and the reported cost is amortized accordingly.
>
> |Component|Qwen2.5-Math-1.5B| |Qwen2.5-3B| |
> |-|-|-|-|-|
> | |Avg Time per Step (s)|Percentage (%)|Avg Time per Step (s)| Percentage (%) |
> |Reference Rollout|37.76|16.78%|61.73|17.36%|
> |Adaptive Difficulty Prediction|1.17|0.46%|1.21|0.34%|
> |Selected Question Rollout|79.27|35.23%|126.94|35.70%|
> |Actor Update|106.95|47.53% |165.69|46.60%|
>
>
> > **Q3 Part 2: Wall-clock efficiency of DOTS (without RR) vs. original GRPO**
>
> **Re.:** Yes, even when using DOTS without RR, **the efficiency advantage in wall-clock time is still present**.
>
> Although DOTS (without RR) introduces a small additional overhead due to reference rollouts and difficulty prediction, it requires **much fewer steps to reach the same final accuracy** as the original GRPO. As a result, the **overall training time is reduced**.
>
> For example, on the Qwen2.5-Math-1.5B + DeepScaleR setting, DOTS reaches the final accuracy of original GRPO **31.23%** faster in terms of wall-clock time. More results can be found in the response to Q5 below.
>
> > **Q5: Add training steps as x-axis in Figure 3 and GPU time in Figure 5**
>
> **Re.:** Thank you for the suggestion. Since figures are not permitted during the rebuttal, we provide the corresponding results in tabular form to address both of your concerns:
>
> (1) **Figure 3 → Now with training steps as x-axis (Table D)**:
>
> (2) **Figure 5 → Now with GPU time as x-axis (Table E)**.
>
> Across both views (training steps and wall-clock time), DOTS & RR and DOTS consistently demonstrate strong performance, confirming the robustness of our advantage **regardless of the presentation format**. We will include the updated figures in the revised appendix.
>
> **Table D: Average accuracy of DOTS+RR (Ours) and original GRPO across various LLM-dataset combinations.**
> | Steps | Qwen2.5-Math-1.5B + MATH | | Qwen2.5-Math-1.5B + DeepScaleR | | Qwen2.5-Math-1.5B + ORZ | | Qwen2.5-3B + DeepScaleR | | Qwen2.5-3B + DeepMath | | Qwen2.5-Math-7B + DeepScaleR | |
> |-|-|-|-|-|-|-|-|-|-|-|-|-|
> ||DOTS+RR|Original GRPO|DOTS+RR|Original GRPO|DOTS+RR|Original GRPO|DOTS+RR|Original GRPO|DOTS+RR| Original GRPO|DOTS+RR|Original GRPO|
> |10|49.03|48.19|48.39|47.99|48.42|48.17|48.93|47.82|47.86|47.66|63.44|61.99|
> |20|50.61|49.82|50.19|49.06|50.72|50.17|49.77|49.09|48.43|48.25|65.07|64.55|
> |30|51.69|51.11|51.52|49.89|52.28|51.46|49.95|49.65|48.86|48.08|66.38|65.76|
> |40|52.42|51.78|52.27|50.81|53.00|52.35|50.25|50.01|49.01|48.29|67.09|66.70|
> |50|52.84|52.27|53.01|51.44|53.75|53.00|50.55|50.31|49.13|48.43|67.54 |67.13|
> |60|53.12|52.66|53.52|51.95|54.15|53.52|50.65|50.47|49.13|48.58|67.91|67.58|
>
> **Table E: Accuracy comparison of DOTS+RR, DOTS (without RR), and original GRPO on Qwen2.5-Math-1.5B in the ablation study. The “Final” row refers to the last evaluation point for each run and the “Time to Achieve original GRPO Final Performance” row refers to the required time to match original GRPO final performance. DOTS can still reduce training time compared to original GRPO. DOTS+RR consistently achieves comparable performance with DOTS while saving the most training time.**
>
> |Time(h)|MATH|||DeepScaleR|||ORZ|||
> |-|-|-|-|-|-|-|-|-|-|
> ||DOTS+RR|DOTS|Original GRPO|DOTS+RR|DOTS|Original GRPO|DOTS+RR|DOTS|Original GRPO|
> |0.50|48.49|47.76|47.75|47.95|47.45|47.65|47.85|47.26|47.36|
> |1.00|50.01|49.03|48.99|49.72|48.86|48.24|50.00|48.94|49.06|
> |1.50|51.03|50.10|50.22|50.90|50.23|49.14|51.42|50.32|50.51|
> |2.00|51.92|51.20|51.11|51.60|51.04|49.71|52.50|51.38|51.34|
> |2.50|52.42|51.71|51.49|52.27|51.60|50.46|53.00|52.13|52.02|
> |3.00|52.84|52.20|52.06|52.86|52.21|50.95|53.75|52.62|52.62|
> |3.50|53.07|52.53|52.27|53.43|52.65|51.44|54.09|53.16|53.00|
> |4.00|–|52.85|52.55|–|53.00|51.76|–|53.56|53.42|
> |Final|53.12|53.28|52.66|53.52|53.52|51.95|54.15|54.12|53.52|
> |Time to Achieve original GRPO Final Performance (h)|2.81|3.64|4.14|2.25|2.91|4.24|2.94|3.81|4.15|
> |Time Saved|**32.1%**|**12.1%**|–|**46.9%**|**31.2%**|–|**29.2%**|**8.3%**|–|
>
> > **Q6: AIME results exclusion**
>
> **Re.:** We initially excluded AIME results because AIME is known to cause instability in evaluation for smaller models due to high variance and limited data size (only 30 questions) [1].
>
> To support this point empirically, we provide **additional experimental results** using various LLM-dataset combinations with the original GRPO. Specifically, we evaluated each of the AIME 24 questions 8 times and computed the average accuracy (avg@8) across training steps.
>
> As shown in the following table, accuracy fluctuates heavily across steps, with no clear trend (results shown every 10 steps). This **high variance across checkpoints** makes it difficult to obtain **reliable evaluation signals** on AIME.
>
> **Table F: Accuracy of original GRPO on AIME 24.**
> |Steps| Qwen2.5-Math-1.5B+DeepScaler|Qwen2.5-3B+DeepScaler|Qwen2.5-Math-7B+DeepScaler|
> |-|-|-|-|
> |10|10.42|7.50|20.42|
> |20|15.83|6.25|24.17|
> |30|15.83|8.75|24.17|
> |40|11.67|9.17|23.33|
> |50|16.67|7.50|25.42|
> |60|13.75|5.42|20.00|
>
> As suggested, we will include AIME evaluation results for all models in the appendix **for reference**, and provide a detailed discussion on why we exclude AIME from the main analysis.
>
> ---
>
> [1] Hochlehnert, Andreas, Hardik Bhatnagar, Vishaal Udandarao, Samuel Albanie, Ameya Prabhu, and Matthias Bethge. "A sober look at progress in language model reasoning: Pitfalls and paths to reproducibility." arXiv preprint arXiv:2504.07086 (2025).

---

> > ### Comment · Reviewer_puy2 · 2025-08-05
> >
> > I would like to thank the authors for providing additional results during the short rebuttal period. My questions are answered. Please include these clarifications and results in the final revised version. Though the improvements are marginal in some experiments, I feel the paper is above the threshold for acceptance. I will raise my score.

---

### Official Review · Reviewer_1Wxg · 2025-07-01

**Clarity:** 3
**Significance:** 2
**Originality:** 2
**Rating:** 4
**Confidence:** 4

**Summary:**

This paper presents a new method for selecting LLM RL training problems via their difficulty level for the current policy. This is done by rolling out the policy on a small core-set of reference problems and estimating problem difficulty for the remainder of the train set using an attention based scoring. Selected rollouts are placed in a replay buffer and used to train the current policy (by additionally modifying the standard GRPO loss accomodate off-policy data). Problems are sampled by the replay buffere reweighted by their likelihood under the policy by which they were generated. Experiments show the method improves sample efficiency by 25%-65%.

**Questions:**

- Why not at least also include reference solutions in the embedding adaptive diffuclty embeddings to better calibrate solution difficulty?
- In a similar direction, why not include an oracle training baseline estimating problem difficulty by rolling out the model for every problem? While this may be costly, it seems possible to maintain slightly off-policy difficulty estimates by reusing training rollouts.

**Ethical Concerns:**

["NO or VERY MINOR ethics concerns only"]

**Limitations:**

Yes

**Quality:**

3

**Strengths And Weaknesses:**

**Strengths:**

- The presentation and layout of the paper is clear and easy to follow.
- The evaluation is done across multiple model scales (1.5B, 3B, 7B) and multiple datasets
- The paper addresses a key bottleneck in the LLM RL training pipeline: rollout generation inefficiency.

**Weaknesses:**

- I'm a a bit skeptical of the adaptive difficulty assessment approach: many questions may be semantically similar in statement but may actually be quite different in difficultly (e.g. consider the simplicity of Goldbach's conjecture). The correlation between estimated difficulty and ground-truth also does not seem to be very strong.
- In the theorem, it is assumed rewards are independent from the gradient ($\nabla_\theta log (o_i | q_i)$). Can the authors motivate why these assumptions are reasonable?

---

> ### Author Rebuttal · Authors · 2025-07-31
>
> We sincerely thank Reviewer 1Wxg for the thoughtful and constructive feedback. In response to the suggestions, we have conducted **additional experiments during the rebuttal period** on (1) providing **empirical evidence** supporting our theoretical analysis assumptions, (2) **including reference solutions** when training the adaptive difficulty prediction framework, and (3) **adding an oracle training baseline** using full rollouts for every question.
>
> Below, we address the questions and concerns point by point.
>
> ---
>
> > **W1: Effectiveness of the adaptive prediction framework & correlation between estimated difficulty and ground-truth**
>
> **Re.:** Thank you for the thoughtful feedback. We agree that in mathematics, surface-level similarity does not always imply similar difficulty. This is precisely **why we do not rely on off-the-shelf embeddings**. Instead, we train an adapter to refine the embedding space to capture difficulty signals rather than surface form. As shown in Table 5 of the ablation study (L148–154, Appendix F.1), this leads to a substantial improvement in prediction alignment with ground-truth adaptive difficulty compared to off-the-shelf alternatives.
>
> While the overall Pearson correlation ranges from 0.7 to 0.8, we emphasize that our goal is not to precisely predict the difficulty of **every question**. Rather, at each training step, we only need to **identify B questions** that are highly likely to be informative (those near 0.5 adaptive difficulty). Our method achieves this effectively: it significantly increases the ratio of effective questions (as shown in Figure 4 on Page 8), and selects about **30–40% more** moderate-difficulty questions compared to random sampling, resulting in substantial data efficiency gains.
>
> > **Q1: Incorporate reference solutions in the embeddings**
>
> **Re.:** Thank you for the suggestion! We conduct an **additional experiment** on the MATH dataset by including reference solutions when constructing the input embeddings. This results in a **slight improvement** in Pearson correlation, indicating that solutions can provide helpful additional signals beyond the problem statement. However, since reference solutions are **not available for all datasets** (e.g., DeepScaleR and ORZ), this approach may have limited applicability. We will include this discussion in the revised version.
>
> **Table: Comparison of average Pearson correlation ($\rho$) between predicted and ground-truth adaptive difficulties.**  Reported as mean ± standard deviation over 60 training steps.
>
> |Model+Dataset|w.o. Solutions | w. Solutions |
> |-|-|-|
> |Qwen2.5-Math-1.5B+MATH| 0.7843 $\pm$ 0.0243 | 0.7940 $\pm$ 0.0229 |
>
>
> > **W2: Independence assumptions in theoretical analysis**
>
> **Re.:** Thank you for your question. We agree that assuming full independence between policy gradients and rewards may be overly strong in practice. In the original proof, we adopted this assumption for simplicity. However, we now provide a more **general and relaxed** formulation **without affecting our main conclusion**.
>
> We now expand the original derivation (L65, Appendix C) to include the covariance correction term $T_2$:
>
> $\mathbb{E}[||g||^2] =\underbrace{\sum\_{i,j=1}^G\mathbb{E}[\hat{A}_i \hat{A}_j] \cdot \mathbb{E}[\nabla\_{\theta}\log \pi\_\theta(o_i \mid q)^\top\nabla\_\theta \log \pi\_\theta(o_j \mid q)]}\_{T_1}$ + $\underbrace{\sum\_{i,j=1}^G \mathrm{Cov} (\hat{A}_i\hat{A}_j,\nabla\_\theta\log \pi\_\theta(o_i \mid q)^\top \nabla\_\theta \log \pi\_\theta(o_j \mid q))}\_{T_2}$.
>
> We denote the first term as the **independence term** and the second as the **covariance correction term**. To relax the independence assumption, we introduce a weak-dependence assumption:
>
> $ \frac{\left|T_2\right|}{T_1}  \ll 1$.
>
> Under this assumption, our original subsequent derivation and final conclusion remains unchanged, as the **independence term continues to dominate**. **This confirms that our previous theoretical conclusion was not incorrect, but rather based on a simplified case that can now be justified under a broader condition.**
>
> * **Empirical evidence:** To support this relaxed assumption **empirically**, we compute the covariance-to-main ratio described above. We randomly sample 512 questions for each of two LLM-dataset combinations, generate 8 rollouts per question, and evaluate the ratio ($ \frac{\left|T_2\right|}{T_1}$). The results are:
>
> – Qwen2.5-Math-1.5B + MATH: 0.081 ± 0.0065
>
> – Qwen2.5-3B + DeepMath: 0.081 ± 0.0051
>
> These low ratios empirically validate the weak-dependence assumption.
>
> We thank the reviewer again for raising this point. We will revise the theoretical section in the final version to explicitly incorporate the relaxed assumption and include the empirical results that support it.
>
> > **Q2: Oracle training baseline with full-rollout difficulty estimates**
>
> **Re.:** Thank you for the insightful suggestion. We implemented such an oracle baseline by performing full rollouts on all questions every $\mu$ steps and selecting $\mu \times B$ questions, prioritizing those with adaptive difficulty near 0.5, for the next $\mu$ training steps. We set $\mu = 6$ to allow the oracle baseline to use approximately **twice the computational budget of DOTS**. To make the comparison fair, we also removed rollout replay from both methods to isolate the effect of difficulty-based selection.
>
> As shown in the table below, this oracle baseline **underperforms our DOTS method**. This is likely because these difficulty scores become increasingly off-policy as training progresses. Increasing the frequency of full rollouts could mitigate this issue, but would incur **even greater computational cost**—recall that running full rollouts every 6 steps already doubles the training cost compared to DOTS.
>
> In contrast, our adaptive difficulty prediction framework strikes a strong balance between **on-policy difficulty estimation** and **computational efficiency**: it **generates on-policy difficulty predictions** with **minimal overhead**, enabling it to better capture evolving training dynamics and deliver stronger performance.
>
>
> **Table: Average accuracy of different methods on Qwen2.5-Math-1.5B+DeepScaleR.**
> | Steps | DOTS (Ours) | Oracle | Original GRPO |
> |-|-|-|-|
> |10| **48.61**| 47.68  | 47.71|
> |20| **50.54**| 49.69  | 49.09|
> |30| **51.74**| 50.84  | 50.16|
> |40| **52.47**| 51.65  | 50.88|
> |50| **53.12**| 52.36  | 51.50|
> |60| **53.60**| 52.69  | 51.93|

---

> > ### Comment · Reviewer_1Wxg · 2025-08-04
> >
> > Thank you for your detailed response and the extra experiments. I will keep my positive score the same.

---

> > > ### Author Response · Authors · 2025-08-04
> > > **Thank you!**
> > >
> > > Thank you again for your thoughtful feedback! We're confident that the additional results and clarifications further strengthen the paper, and we truly appreciate your engagement.
> > >
> > > If there’s anything else we can clarify to support your evaluation, please don’t hesitate to let us know — we’re committed to continuously improving the quality of our work.

---

### Official Review · Reviewer_9x3o · 2025-07-02

**Clarity:** 3
**Significance:** 3
**Originality:** 2
**Rating:** 4
**Confidence:** 4

**Summary:**

This paper proposes a method to improve data efficiency in reinforcement learning (RL) fine-tuning of large language models (LLMs), primarily through two techniques: Difficulty-targeted Online Data Selection (DOTS) and Rollout Replay (RR) . The authors validate the effectiveness of their approach both theoretically and experimentally, conducting extensive evaluations across multiple LLM-dataset combinations. The results show that the proposed method reduces training time by 25% to 65% while maintaining performance.

Overall, the paper is well-structured, clearly written, and presents a thorough experimental design with strong practical value. Especially given the high computational cost of current RL training methods, this work offers an efficient and reproducible solution.

**Questions:**

--Enhance Novelty Explanation:
Clarify the core differences between DOTS and automatic curriculum learning methods, emphasizing the innovation of using an attention mechanism to achieve zero-shot difficulty transfer.
--Supplement Empirical Validation of the Independence Assumption:
Conduct empirical tests of the independence assumption in Theorem 1, such as binning statistics of gradient norm distributions across different success rate intervals p .
--Refine Method Details:
Clearly specify the architecture of the MLP prediction framework, and discuss how well the assumptions in Theorem 1 hold in practical policy gradient settings.
--Error Analysis by Question Type and Dynamic Reference Set Expansion:
Break down prediction errors by question type, and explore mechanisms for dynamically expanding the reference set. Include additional experiments—such as evaluating on non-mathematical tasks (e.g., AlpacaEval)—to validate generalization.

**Ethical Concerns:**

["NO or VERY MINOR ethics concerns only"]

**Final Justification:**

The author's response has addressed my concerns, and I keep my initial rating of Borderline Accept.

**Limitations:**

Yes.

**Quality:**

3

**Strengths And Weaknesses:**

Strengths:
--Clear problem definition:
The paper precisely identifies the core issue of low data efficiency in LLM reinforcement fine-tuning and quantifies the severity of the problem using computational cost analysis (e.g., 3,800 A100 GPU hours / $4,500). The motivation is clear and highly relevant to real-world applications.
--Comprehensive experimental design:
The experiments cover six LLM-dataset combinations (ranging from 1.5B to 7B parameters) and are evaluated on widely used mathematical reasoning datasets such as MATH and DeepMath. Multiple evaluation metrics are employed, providing solid empirical support for the claims.
--Strong theoretical foundation:
Appendix provides a complete proof of Theorem 1 , which states that gradient magnitude is maximized when the reward success rate is 50%. This theorem underpins the difficulty selection strategy and is supported by rigorous mathematical derivation.

Weaknesses:
--Incremental innovation:
While DOTS and RR are effective, they represent incremental improvements over existing methods. Difficulty-based curriculum learning has been studied extensively in RL, and the paper does not sufficiently clarify the fundamental differences between DOTS and classical curriculum learning approaches. Similarly, RR is essentially a variant of experience replay, adapted for GRPO, but its technical contribution remains a straightforward extension rather than a novel breakthrough.
--Assumption mismatch in Theorem 1:
Theoretical result Theorem 1 assumes independent and identically distributed (i.i.d.) Bernoulli rewards and that the policy gradient is independent of the reward distribution. However, in LLM generation tasks, token predictions are inherently sequential and interdependent—early errors can propagate and affect later tokens and final rewards. Thus, the i.i.d. assumption may not hold, potentially undermining the validity of the theorem in practical settings.
--Missing implementation details of the prediction framework:
The lightweight MLP adapter used for adaptive difficulty prediction lacks detailed architectural description (e.g., number of layers, activation functions). Additionally, it is unclear how the training data for the adapter is constructed and whether it is independent of the RL training process.
--Potential failure cases in difficulty prediction:
The prediction framework estimates adaptive difficulty via embedding similarity interpolation. However, in domains like mathematics, surface-level similarity does not necessarily imply similar difficulty levels. For example, "taking derivatives" and "computing integrals" may have similar embeddings but vastly different difficulties. Furthermore, the reference set (only 256 samples) may lack sufficient coverage, leading to unreliable predictions for unseen question types.

---

> ### Author Rebuttal · Authors · 2025-07-31
>
> We sincerely thank Reviewer 9x3o for the thoughtful and constructive feedback. In response to the suggestions, we have conducted **additional experiments during the rebuttal period** on (1) providing **empirical evidence** supporting our theoretical analysis assumption and (2) **increasing the reference set size** from 256 to 512 to demonstrate that our current choice is sufficient.
>
> Below, we address the questions and concerns point by point.
>
> ---
>
> > **W1 & Q1:Clarification on innovation**
>
> **Re.:** We would like to clarify the novelty of our contributions. Prior work on difficulty-based curriculum learning either relies on *external difficulty labels* [1,2], which we show in §5.4 to be less effective than our method, or computes *adaptive difficulty* but requires full rollouts for every question [3], making them computationally impractical. In contrast, we propose a **novel attention-based adaptive difficulty prediction framework** that **efficiently and accurately estimates adaptive difficulty without full rollouts**. We also provide Proposition 1 to theoretically justify the benefit of selecting questions with difficulty near 0.5. Together, these innovations enable scalable and principled online data selection for LLM RL fine-tuning, which prior methods cannot achieve due to either limited effectiveness or prohibitive cost.
>
> Regarding RR, while experience replay is common in off-policy RL, it has not, to our knowledge, been applied to LLM RL fine-tuning; we **bridge this gap** by adapting it with a modified GRPO loss to enable stable and efficient rollout reuse.
>
> Finally, our method improves data efficiency of LLM RL fine-tuning from two **complementary angles**: DOTS reduces the number of steps needed, while RR lowers the cost per step. This unified approach to optimizing both axes has not been explored in prior work.
>
> > **W2 & Q2: Independence assumptions in theoretical analysis**
>
> **Re.:** Thank you for your question. It involves two parts: (1) the i.i.d. assumption on Bernoulli rewards and (2) the assumption that the policy gradient is independent of the reward. We address both below.
>
> (1) **i.i.d. reward assumption is reasonable and standard**
>
> This assumption holds in our setting because **given the same prompt**, each response is **randomly sampled** from the **same policy**, and the reward is then computed by a deterministic rule-based verifier. As a result, the only source of randomness lies in the policy’s stochastic generation process, and the rewards are i.i.d. from the same Bernoulli (p) distribution. This assumption is commonly used in prior work [4,5].
>
> (2) **Independence between gradient and reward can be relaxed without affecting the main conclusion & empirical evidence to support this relaxed assumption**
>
> We agree that assuming full independence between policy gradients and rewards may be overly strong in practice. In the original proof, we adopted this assumption for simplicity. However, we now provide a more **general and relaxed** formulation **without affecting our main conclusion**.
>
> We now expand the original derivation (L65, Appendix C) to include the covariance correction term:
>
> $\mathbb{E}[||g||^2] =\underbrace{\sum\_{i,j=1}^G\mathbb{E}[\hat{A}_i \hat{A}_j] \cdot \mathbb{E}[\nabla\_{\theta}\log \pi\_\theta(o_i \mid q)^\top\nabla\_\theta \log \pi\_\theta(o_j \mid q)]}\_{T_1}$ + $\underbrace{\sum\_{i,j=1}^G \mathrm{Cov} (\hat{A}_i\hat{A}_j,\nabla\_\theta\log \pi\_\theta(o_i \mid q)^\top \nabla\_\theta \log \pi\_\theta(o_j \mid q))}\_{T_2}$.
>
> We denote the first term as the **independence term** and the second as the **covariance correction term**. To relax the independence assumption, we introduce a weak-dependence assumption:
>
> $ \frac{\left|T_2\right|}{T_1}  \ll 1$.
>
> Under this assumption, our original subsequent derivation and final conclusion remains unchanged, as the **independence term continues to dominate**. **This confirms that our previous theoretical conclusion was not incorrect, but rather based on a simplified case that can now be justified under a broader condition.**
>
> + **Empirical evidence:** To support this relaxed assumption **empirically**, we compute the covariance-to-main ratio described above. We randomly sample 512 questions for each of two LLM-dataset combinations, generate 8 rollouts per question, and evaluate the ratio. The results are:
>
> – Qwen2.5-Math-1.5B + MATH: 0.081 ± 0.0065
>
> – Qwen2.5-3B + DeepMath: 0.081 ± 0.0051
>
> These low ratios empirically validate the weak-dependence assumption.
>
> We thank the reviewer again for raising this point. We will revise the theoretical section in the final version to explicitly incorporate the relaxed assumption and include the empirical results that support it.
>
> > **W3 Part 1 & Q3: Implementation details of adaptive difficulty framework**
>
> **Re.:** The implementation details of the adaptive difficulty framework are provided in L81-95 in Appendix D.1. Specifically, the adapter is a GELU-activated MLP with three hidden layers, each containing 896 units and a dropout rate of 0.1. A LayerNorm is applied to the projection output to stabilize training.
>
> > **W3 Part 2: The construction of training data for the adapter**
>
> The construction process is detailed in Appendix D.1 (L87–93). Each training instance consists of a query question, a reference set with known difficulty scores, and a ground-truth adaptive difficulty label. To build this dataset, we sample questions from the math datasets and use a fixed set of **external LLMs (disjoint from the policy model)** to generate responses and compute adaptive difficulty scores, which serve as supervision signals for training the adapter.
>
> The adapter training **is independent of the RL process**, as the policy model trained during RL is excluded at this stage, and the adapter is trained before RL fine-tuning begins. This ensures that **no information from the policy model is leaked**.
>
>
> > **W4 & Q4 Part 1: Potential failure case of adaptive prediction framework & Coverage of Reference Set**
>
> **Re.:** We agree that in mathematics, surface-level similarity does not always imply similar difficulty. This is precisely **why we do not rely on off-the-shelf embeddings**. Instead, we train an adapter to refine the embedding space to capture difficulty signals rather than surface form. As shown in Table 5 of the ablation study (L148–154, Appendix F.1), this leads to a substantial improvement in prediction alignment with ground-truth adaptive difficulty compared to off-the-shelf alternatives.
>
> To **demonstrate that a reference set size of 256 is sufficient**, we conduct **additional experiments** increasing the size to 512. As shown in the table below, the improvement in adaptive difficulty prediction is **marginal** (typically 1–2%). This suggests that increasing the reference set beyond 256 offers diminishing returns in prediction performance while incurring higher computational cost.
>
> Finally, we agree that dynamically adjusting the reference set size based on question types is an interesting direction. We leave this exploration to future work.
>
> **Table: Average Pearson correlation ($\rho$) between predicted and ground-truth adaptive difficulties under different reference set sizes.**
>
> |Model|Dataset|Ref Size = 256|Ref Size = 512|
> |-|-|-|-|
> |Qwen2.5-Math-1.5B|DeepScaleR|0.7244 $\pm$ 0.0318 |0.7434 $\pm$ 0.0365|
> |Qwen2.5-3B|DeepScaleR |0.7789 $\pm$ 0.0191| 0.7916 $\pm$ 0.0193|
> |Qwen2.5-Math-7B|DeepScaleR|0.7076 $\pm$ 0.0195| 0.7140 $\pm$ 0.0431|
>
> > **Q4 Part 2: Generalization to non-math domains**
>
> **Re.:** We would greatly appreciate clarification on which aspect of generalization you are referring to:
>
> (1) If the concern is about the adaptive difficulty prediction framework, we note that cross-domain generalization is not required in our setting. Instead, for each domain, we can train a separate MLP adapter, which is both **justified and practical**.
>
> - **Justification:** This adapter is trained **specifically for the data domain used in the subsequent LLM RL fine-tuning**, which is **known in advance**. **If data from a particular domain will be used during RL fine-tuning, it is naturally included in the adapter training data.** Therefore, there is no need to generalize to out-of-domain data. This setting is fundamentally different from standard supervised learning, where models are expected to perform well on unseen distributions.
>
> - **Practicality:** The training cost is low, taking about 11.5 minutes on a single NVIDIA A100 GPU. This makes per-domain retraining entirely manageable in practice.
>
> (2) If the question instead concerns the generalization of our full data-efficient RL fine-tuning pipeline beyond math, we have **conducted an additional experiment in the science domain** using the SCP-25K dataset. Evaluation results on the science subsets of MMLU confirm that our method continues to **significantly improve data efficiency**, supporting its **broader applicability**. Please refer to our response to Q2 & L1 of reviewer NvgW for detailed results due to space limits here.
>
> ---
>
> [1] Lee, Bruce W., Hyunsoo Cho, and Kang Min Yoo. "Instruction Tuning with Human Curriculum." In Findings of the Association for Computational Linguistics: NAACL 2024, pp. 1281-1309. 2024.
>
> [2] Team, Kimi, Angang Du, Bofei Gao, Bowei Xing, Changjiu Jiang, Cheng Chen, Cheng Li et al. "Kimi k1. 5: Scaling reinforcement learning with llms." arXiv preprint arXiv:2501.12599 (2025).
>
> [3] Bae, Sanghwan, Jiwoo Hong, Min Young Lee, Hanbyul Kim, JeongYeon Nam, and Donghyun Kwak. "Online difficulty filtering for reasoning oriented reinforcement learning." arXiv preprint arXiv:2504.03380 (2025).
>
> [4] Vojnovic, Milan, and Se-Young Yun. "What is the Alignment Objective of GRPO?." arXiv preprint arXiv:2502.18548 (2025).
>
> [5] Shi, Taiwei, Yiyang Wu, Linxin Song, Tianyi Zhou, and Jieyu Zhao. "Efficient reinforcement finetuning via adaptive curriculum learning." arXiv preprint arXiv:2504.05520 (2025).

---

> > ### Author Response · Authors · 2025-08-07
> > **Thank you!**
> >
> > Dear Reviewer,
> >
> > Thank you again for your thoughtful review. We have carefully addressed each of your comments and hope that our responses have fully clarified the issues you raised.
> >
> > As we have not yet seen a reply, we wanted to kindly check if there are any remaining questions or concerns we can help clarify before the discussion period ends.
> >
> > Thank you again for your time and efforts in reviewing our submission.
> >
> > Best regards,
> >
> > The Authors

---

### Note · Authors · 2025-08-16

Dear AC and Reviewers,

We sincerely thank you for the constructive feedback and active engagement throughout the review process. We are encouraged that all reviewers recognized the strengths of our work, including:

- a well-motivated approach addressing a key bottleneck in LLM RL fine-tuning to improve data efficiency,

- comprehensive evaluations across multiple model scales and datasets,

- demonstrated substantial time savings.

During the rebuttal period, besides clarifying key points and providing empirical evidence supporting our theoretical assumptions, we conducted **comprehensive additional experiments** to address major concerns:

- **Statistical significance**: We added error bars for three LLM-dataset combinations by repeating both DOTS+RR and original GRPO three times. Our improvements over original GRPO are confirmed to be statistically significant, with **nearly all p-values below 0.05**.

-  **Extending training horizon**: We extended training to 100 steps to observe convergence behavior, which consistently demonstrates the superior performance of our method.

- **Applications beyond mathematics**: New experiment in the Science domain validates that DOTS+RR outperforms original GRPO in **non-math** settings.

We are grateful that the reviewers for acknowledging that our additional results addressed their concerns, and that **all reviewers provided positive assessments** after the rebuttal period. We will include all additional experiments in the final revision to further strengthen the paper.

---

### Decision · Program_Chairs · 2025-09-17

**Decision:**

Accept (poster)

**Comment:**

This paper proposes a method to improve data efficiency in reinforcement learning (RL) fine-tuning of LLMs. All reviewers and I agree that the paper is well-written and well-motivated. The experiments are finally comprehensive, while some reviewers raised concerns about experiments in the initial phase.

During the rebuttal phase, the authors made great efforts and supplemented additional experiments, so most reviewers were satisfied with the response and raised their scores to BORDERLINE ACCEPT.

After reading the paper, I'm also inclined to accept this paper. The authors should revise the paper in the camera-ready version according to the suggestions given by reviewers